# Immune cell dynamics deconvoluted by single-cell RNA sequencing in normothermic machine perfusion of the liver

T. Hautz[1,6], S. Salcher [2,6], M. Fodor[1,6], G. Sturm [3], S. Ebner[1], A. Mair[2], M. Trebo [2], G. Untergasser[2,4], S. Sopper[2], B. Cardini[1], A. Martowicz [2,4], J. Hofmann[1], S. Daum [2], M. Kalb[2], T. Resch[1], F. Krendl [1], A. Weissenbacher [1], G. Otarashvili[1], P. Obrist[4], B. Zelger[5], D. Öfner [1], Z. Trajanoski [3], J. Troppmair [1], R. Oberhuber[1], A. Pircher [2], D. Wolf[2,7] ✉ & S. Schneeberger [1,7] ✉

Normothermic machine perfusion (NMP) has emerged as an innovative organ preservation technique. Developing an understanding for the donor organ immune cell composition and its dynamic changes during NMP is essential. We aimed for a comprehensive characterization of immune cell (sub)populations, cell trafficking and cytokine release during liver NMP. Single-cell transcriptome profiling of human donor livers prior to, during NMP and after transplantation shows an abundance of CXC chemokine receptor $1^+/2^+$ ($CXCR1^+/CXCR2^+$) neutrophils, which significantly decreased during NMP. This is paralleled by a large efflux of passenger leukocytes with neutrophil predominance in the perfusate. During NMP, neutrophils shift from a pro-inflammatory state towards an aged/chronically activated/exhausted phenotype, while anti-inflammatory/tolerogenic monocytes/macrophages are increased. We herein describe the dynamics of the immune cell repertoire, phenotypic immune cell shifts and a dominance of neutrophils during liver NMP, which potentially contribute to the inflammatory response. Our findings may serve as resource to initiate future immune-interventional studies.

Liver transplantation (LT) is the sole definitive treatment for end-stage liver disease[1–3]. Organ shortage remains a major limiting factor with the demand far exceeding supply. The need for better preservation technologies in conjunction with higher utilization rates in organs from extended criteria donors (ECD) fed into the development of normothermic machine perfusion (NMP). Thereby, an organ is continuously perfused under near to physiologic conditions at 37 °C with oxygenated, heparinized erythrocyte concentrates supplemented with nutrition and antibiotics and in a closed sterile system. NMP enables for a comprehensive assessment of organ quality and function during ex vivo preservation and may serve as a platform for extra-corporal organ reconditioning, treatment, and repair[2,4–9].

Many hundreds of organs have been machine-perfused with success and the short-term results indicate that NMP could help decrease

[1]Department of Visceral, Transplant and Thoracic Surgery, Center of Operative Medicine, organLife Laboratory and D. Swarovski Research Laboratory, Medical University of Innsbruck, Innsbruck, Austria. [2]Department of Internal Medicine V, Hematology and Oncology, Comprehensive Cancer Center Innsbruck (CCCI), Medical University of Innsbruck, Innsbruck, Austria. [3]Institute of Bioinformatics, Biocenter, Medical University of Innsbruck, Innsbruck, Austria. [4]Tyrolpath Obrist Brunhuber GmbH, Zams, Austria. [5]Institute of Pathology, Neuropathology and Molecular Pathology, Medical University of Innsbruck, Innsbruck, Austria. [6]These authors contributed equally: T. Hautz, S. Salcher, M. Fodor. [7]These authors jointly supervised this work: D. Wolf, S. Schneeberger. ✉e-mail: dominik.wolf@i-med.ac.at; stefan.schneeberger@i-med.ac.at

**Table 1 | Overview of donor and preservation data of the study population**

| | |
|---|---|
| Total livers included | $n = 34$ |
| Gender | Female $n = 18$, male $n = 16$ |
| Donor age[a] (years) | 62 (52.25–70) |
| Donor type | DBD $n = 25$, DCD $n = 9$ |
| ECD | Yes $n = 28$, no $n = 6$ |
| CIT[a] (h) | 5.91 (4.70–7.69) |
| NMP time[a] (h) | 19.13 (11.56–21.64) |
| Total preservation time[a] (h) | 25.45 (18.08–27.62) |
| Transplantation after NMP | Yes $n = 26$, no $n = 8$ |
| Livers included for scRNASeq | $n = 8$ |
| Gender | Female $n = 5$, male $n = 3$ |
| Donor age[a] (years) | 66 (53 – 70.5) |
| Donor type | DBD $n = 7$, DCD $n = 1$ |
| ECD | Yes $n = 6$, no $n = 2$ |
| CIT[a] (h) | 5.31 (4.41–5.8) |
| NMP time[a] (h) | 18.08 (13.15–20.08) |
| Total preservation time[a] (h) | 22.46 (19.55–25.66) |
| Transplantation after NMP | Yes $n = 6$, no $n = 2$ |
| Livers included for perfusate analysis | $n = 26$ |
| Gender | Female $n = 15$, male $n = 11$ |
| Donor age[a] (years) | 60.5 (52.25–67.50) |
| Donor type | DBD $n = 18$, DCD $n = 8$ |
| ECD | Yes $n = 22$, no $n = 4$ |
| CIT[a] (h) | 6.13 (4.94–7.77) |
| NMP time[a] (h) | 19.88 (11.02–21.64) |
| Total preservation time[a] (h) | 26.07 (17.65–28.86) |
| Transplantation after NMP | Yes $n = 18$, no $n = 8$ |

*n*: number, *DCD* donation after circulatory death, *DBD* donation after brain death, *ECD* extended criteria donor, *CIT* cold ischemia time, *NMP* normothermic machine perfusion.
[a]Values are median (IQR).

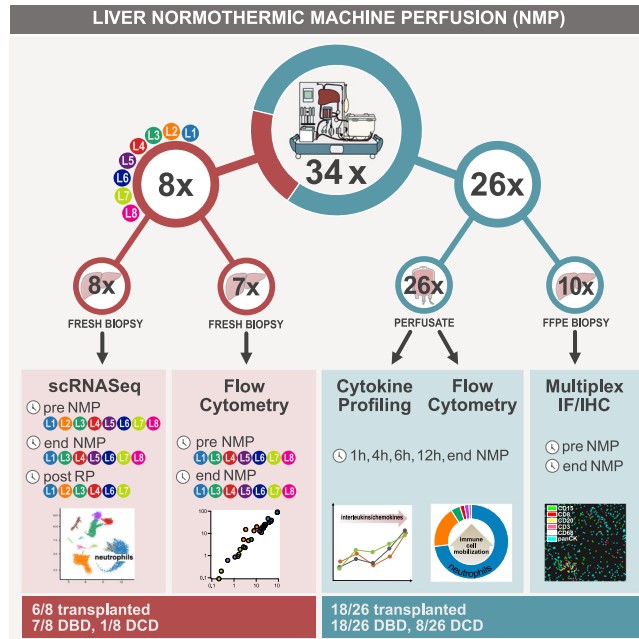

**Fig. 1 | Overview of study.** While a total of 34 donor livers were included in this study and subjected to NMP, eight out of 34 donor livers were randomly selected for scRNASeq, and 26 out of 34 for perfusate sampling. In addition, tissue samples were collected in all 34 livers to assess and validate findings on protein level. Information is provided on the sampling time points, analysis, donor as well as transplantation status of the organs. FFPE fresh frozen paraffin-embedded, NMP normothermic machine perfusion, RP reperfusion, IF immunofluorescence, IHC immunohistochemistry, DBD donor after brain death, DCD donor after circulatory death.

In this work, deep immune cell mapping shows the predominance of neutrophils in human donor livers, which are mobilized into the perfusate early during liver NMP. The assessment of the phenotype and the dynamics of the hepatic immune cell repertoire together with the corresponding cell-to-cell communication patterns enhance our understanding of the immunology of normothermically machine-perfused livers.

## Results

### Study cohort and performance during NMP

An overview of the overall study population is presented in Table 1 (individual data are given in Supplementary Table 1). Detailed information on study livers and analysis is provided as workflow scheme in Fig. 1. The decision to apply NMP was based on one or a combination of the following indications: (D) presumed low organ quality, (R) surgically complex recipient, and/or (L) logistics (Source Data file). NMP time depended on the time required for assessment and targeted time for surgery[2,4,25]. Median NMP time was 19.13 h (h) (interquartile range (IQR) 11.56–21.64). Median peak of aspartate transaminase (AST) and alanine transaminase (ALT) were 6410 U/l (IQR 2422–13,696) and 3194 U/l (IQR 1402–7791), the median lactate level at 6 h was 12 mmol/l (IQR 18–180). The decision to transplant or discard a liver was based on perfusate parameters as previously defined[2,4,25]. Following pre-defined criteria (see methods), 24 livers were transplanted after NMP, while ten were declined because of the inability to maintain physiological pH values, inadequate lactate degradation, and high AST/ALT perfusate levels.

### Recipient characteristics and post-operative outcome

Median recipient model for end-stage liver disease (MELD) and balance of risk (BAR) score were 14 (IQR 11–18) and 7 (IQR 7–10). The median recipient age was 60 years (IQR 55–68). The median intensive care unit

organ discard rates and serve a proper selection of organs suitable for transplantation[4,10–12]. However, while rough organ function is readily measurable, only little is known about the immune status of an organ and its tissue-resident leukocytes during NMP. In experimental studies of ex vivo lung and kidney perfusion, a great number of passenger leukocytes are mobilized and extravasate into the perfusate, thereby affecting graft immunogenicity[13,14]. Data suggestive for mobilization of passenger leukocytes into the perfusate have also been published in liver NMP[15]. Considering the inflammatory capability and the magnitude of liver-intrinsic immune cells, a deeper understanding of the impact of NMP on the hepatic immune cell status and its dynamics during NMP is critically important. Herein, detailed passenger leukocyte mapping using whole transcriptome single-cell RNA sequencing (scRNASeq) technology[16] was considered to significantly enhance our understanding of molecular mechanisms including inflammatory circuits during liver NMP. In recent years, scRNASeq studies aimed to assess the cellular heterogeneity of human livers under various conditions, including ischemia/reperfusion injury (IRI)[17–24]. In-depth serial immune cell mapping of eight human donor livers prior to and at the end of NMP was applied to specifically investigate the source of immunomodulatory cytokines/chemokines and the dynamics of immune activation at the single-cell level. We validated and visualized our findings by multiplex immunofluorescent (IF) staining in liver biopsies. The corresponding immune cell mobilization was characterized by phenotyping in serial perfusate samples collected during 26 human liver NMPs, as well as cytokine profiling and quantification.

**Table 2 | Clinical recipients and post-operative outcome**

| | |
|---|---|
| Age (y)[a] | 60 (55–68) |
| Sex | |
| Male (*n*) | 19 (79.2) |
| Female (*n*) | 5 (20.8) |
| BMI (kg/m²)[a] | 26 (22–28) |
| MELD[a] | 14 (11–18) |
| BAR score[a] | 7 (7–10) |
| Total hospital stay (d)[a] | 21 (15–30) |
| ICU stay (d)[a] | 5 (3–9) |
| Early allograft dysfunction (n) | 9 (37) |
| Clavien Dindo > 3 | 11 (46) |
| Biliary complications (n) | 13 (54) |
| ≤Day 30 (*n*) | 9 (37) |
| >Day 30 (*n*) | 4 (17) |
| Biliary leakage (*n*) | 7 (29) |
| Anastomotic stricture (*n*) | 6 (25) |
| Arterial complication (n) | 5 (21) |
| Patient survival (d)[a] | 358 (83–591) |
| Graft survival (d)[a] | 358 (83–591) |
| 30 days patient survival (n) | 20 (83) |
| 30 days graft survival (n) | 20 (83) |
| Patient death (n) | 5 (21) |
| Follow-up (d)[a] | 143 (35–405) |

*BMI* body mass index; *ICU* intensive care unit; *MELD* model for end-stage liver disease; *BAR* balance of Risk; *y* years; *d* days; *n* number.
Values from *n* = 24 patients given in parentheses are percentages unless indicated otherwise.
[a]Values are median (IQR).

(ICU) and total hospital stay were 5 (IQR 3–9) and 21 days (IQR 15–30). Nine patients (37%) developed early allograft dysfunction (EAD). Clavien-Dindo grade >3 complications occurred in 11 (46%) patients. Three patients died from fungal sepsis, one of clostridium difficile sepsis, and one following cholangiosepsis. Death-censored graft survival was 100% (Table 2).

## Deep immune cell mapping shows predominance of neutrophils in human donor livers

As a first step, the overall immune cell repertoire of eight donor livers subjected to NMP was characterized pre-NMP (T0), at the end of NMP (T1) and after reperfusion (T2) using scRNASeq profiling (Fig. 2A). We annotated most relevant cell populations using unsupervised clustering (Fig. 2B). Our analyses describe the cellular composition and gene expression dynamics of all liver cell types composing the parenchyma including liver inflammatory cells. While the entire data set is made available and gene expression phenotype and dynamics of constituent cells are interesting and relevant, the focus of this study was the dynamic changes of hepatic inflammatory cells during ex vivo perfusion of human donor livers.

Marker genes were analyzed to annotate specific cell types, such as neutrophils (*FCGR3B*), monocytes/macrophages (*CD68*), CD3⁺ T cells (*CD3E*), CD4⁺ T cells (*CD4*), CD8⁺ T cells (*CD8A*), regulatory T cells (Tregs) (*FOXP3*), Natural killer (NK) cells (*NKG7*), dendritic cells (*FLT3*), progenitor cells (*CD24*), B cells (*CD79A*), plasma cells (*JCHAIN*), hepatocytes (*ALB*), endothelial cells (*FLT1*) and cholangiocytes (*KRT19*; Fig. 2D, E, Supplementary Fig. 2). To ensure robust data interpretation, we focused on marker gene signatures to define distinctive cell types, which corroborated the validity of our annotation process (Source Data file).

While some inter-patient heterogeneity was seen, the overall cellular composition was comparable between organs (Fig. 2C,

Supplementary Fig. 3A, B). Neutrophils were identified as the major immune cell cluster in all patients (48.7%; 57,564 cells), and the monocyte/macrophage linage as the second most overall dominant immune cell type within the liver (7.75%; 18,682 cells; Fig. 2C; individual patients see Source Data file). Concordantly, immunohistochemistry (IHC) showed neutrophils (CD15) and monocytes/macrophages (CD68) as dominant resident leukocyte populations in liver tissue (Supplementary Fig. 4). The proportion of leukocytes identified by scRNASeq was validated by flow cytometry in the corresponding samples (Fig. 2F).

Neutrophils were defined by an exceptionally low mRNA content and thus by a relatively low number of detected transcripts per cell (Fig. 2G, H; additional QC metrics are shown in Supplementary Fig. 3D, E). We have recently demonstrated, that—compared to other scRNA-Seq platforms—the BD Rhapsody workflow captures a notably high number of mRNA molecules per cell and may thus be particularly well suitable to depict low mRNA content cells[26]. Hence, due to the relatively high number of mRNA molecules captured per cell, neutrophils with a relatively low mRNA content were identified in our scRNASeq dataset (Fig. 2H; mean nCount_RNA: 4106; mean nCount_RNA in neutrophils: 2050) but not in previously published data sets.

In summary, the immune cell atlas is based on a total of 21 biopsy specimens collected from eight human donor livers prior, at the end of NMP as well as after reperfusion. Our analyses deconvolute the cellular composition and depict previously unrecognized neutrophils at single-cell resolution.

## NMP induces neutrophil decrease over time

In order to investigate the impact of NMP on the liver immune cell landscape we assessed differences in scRNASeq profiles of serial biopsy samples taken pre-NMP (T0) and at the end of NMP (T1) (Fig. 3A; cell numbers of each cell type at T0 and T1 are shown in the Source Data file).

Most striking, the proportion of neutrophils decreased with perfusion time, whereas the proportion of monocytes/macrophages, T cells, and B cells changed only marginally during NMP (Fig. 3B). Using multiplex IF staining of various immune cells in biopsies collected from 10 additional livers confirmed these findings: NMP did not markedly affect the number and distribution of monocytes/macrophages (CD68), T cells (CD3 and CD8), and B cells (CD20), while the absolute number of neutrophils (CD15) significantly diminished ($p < 0.001$) over time (Fig. 3D–F), despite neutrophils remaining prominent throughout the perfusion period (Fig. 3B, D–F).

Interestingly, NMP induced a notable shift in the transcriptomic profile of neutrophils and monocytes/macrophages (Fig. 3C), while the cellular stress level defined by the relative amount of detected mitochondrial transcripts (%MT, Supplementary Fig. 5A) and the gene expression levels of a set of important stress- and apoptosis-related transcripts (Supplementary Fig. 5B) were not markedly affected by NMP. In summary, NMP significantly decreases hepatic neutrophils in quantity and proportion over perfusion time, while other main immune cell types are not markedly affected.

## NMP shifts hepatic neutrophils from a activated pro-inflammatory towards an aged/chronically activated/exhausted phenotype

We then assessed the transcriptomic signature of the neutrophils comprising the largest intrahepatic immune cell population, and annotated subpopulations to investigate the impact of NMP on gene expression and phenotype changes in greater detail. Figure 4A depicts known marker genes in neutrophils, including *IFITM2, CSF3R, FPR1, FCGR3B, VNN2, GOS2, CXCR2*, and *SOD2*. *CXCR2* is of particular interest, as it is predominantly expressed in the neutrophil cluster (Fig. 4B, C; individual patient and biopsy data are shown in Supplementary Fig. 6A). We observed a comparable expression pattern for *CXCR1*

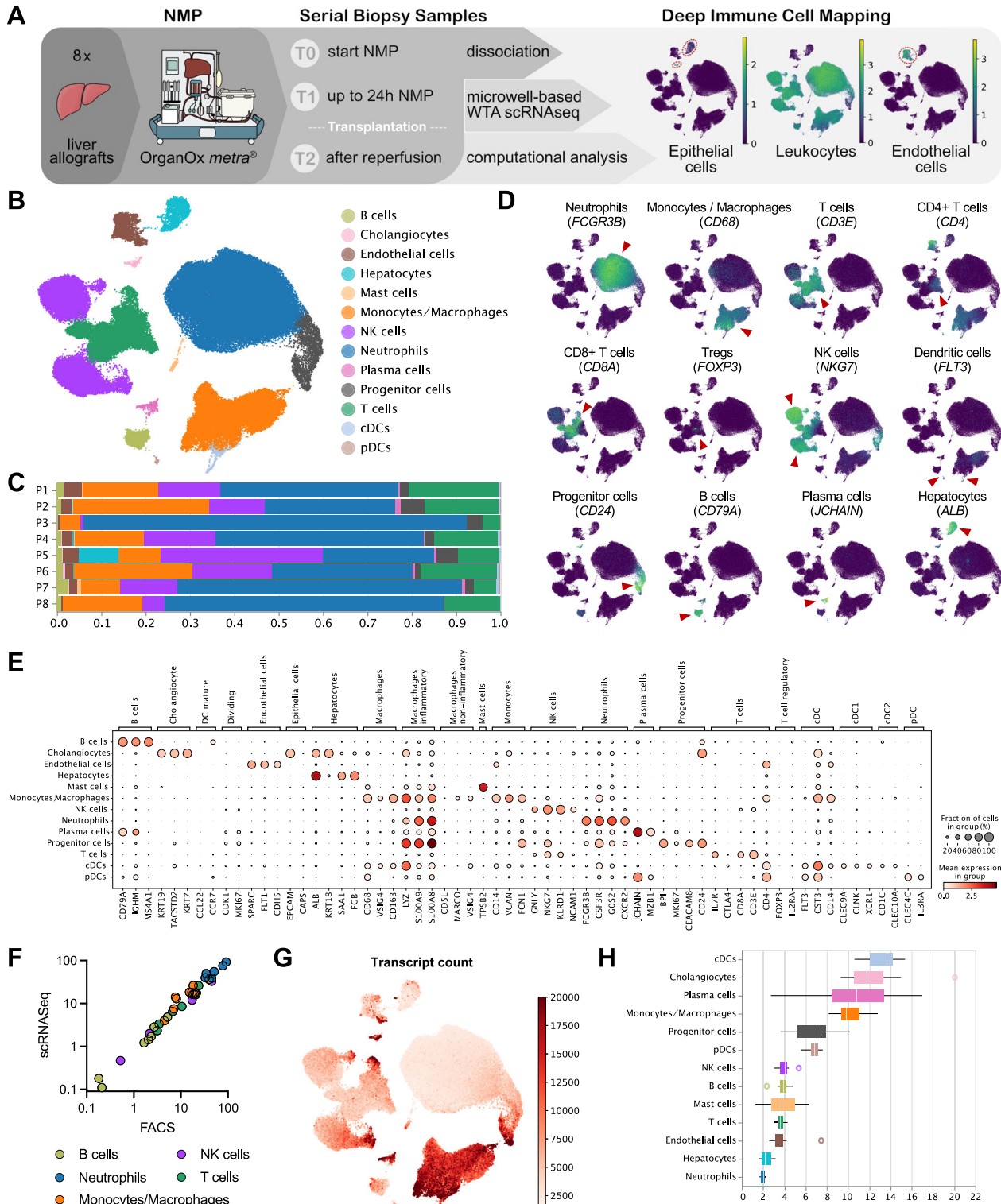

**Fig. 2 | scRNASeq profiling of liver allografts. A** Overview of the scRNASeq workflow. The proportions of epithelial cells (*KRT18*), leukocytes (*PTPRC*/CD45), and endothelial cells (*SPARC*) in the obtained scRNASeq dataset are indicated. **B** Uniform manifold approximation and projection (UMAP) plot of 118,448 single cells, color-coded by cell type. **C** Relative cell-type composition in liver tissues from eight individual patients. **D** UMAP plots, color-coded for the expression of indicated cell-type specific marker genes (red arrowheads). **E** Gene-expression levels of cell-type specific markers. **F** The proportion of leukocyte populations in liver tissue as determined by scRNASeq *vs* flow cytometry. **G** UMAP plot colored by the number of transcripts per cell. The color scale is clipped at 20,000. **H** Boxplot of the transcript count per cell-type. The values denote the average per patient (*n* = 8). The central line denotes the median. Boxes represent the interquartile range (IQR) of the data, whiskers extend to the most extreme data points within 1.5 times the IQR.

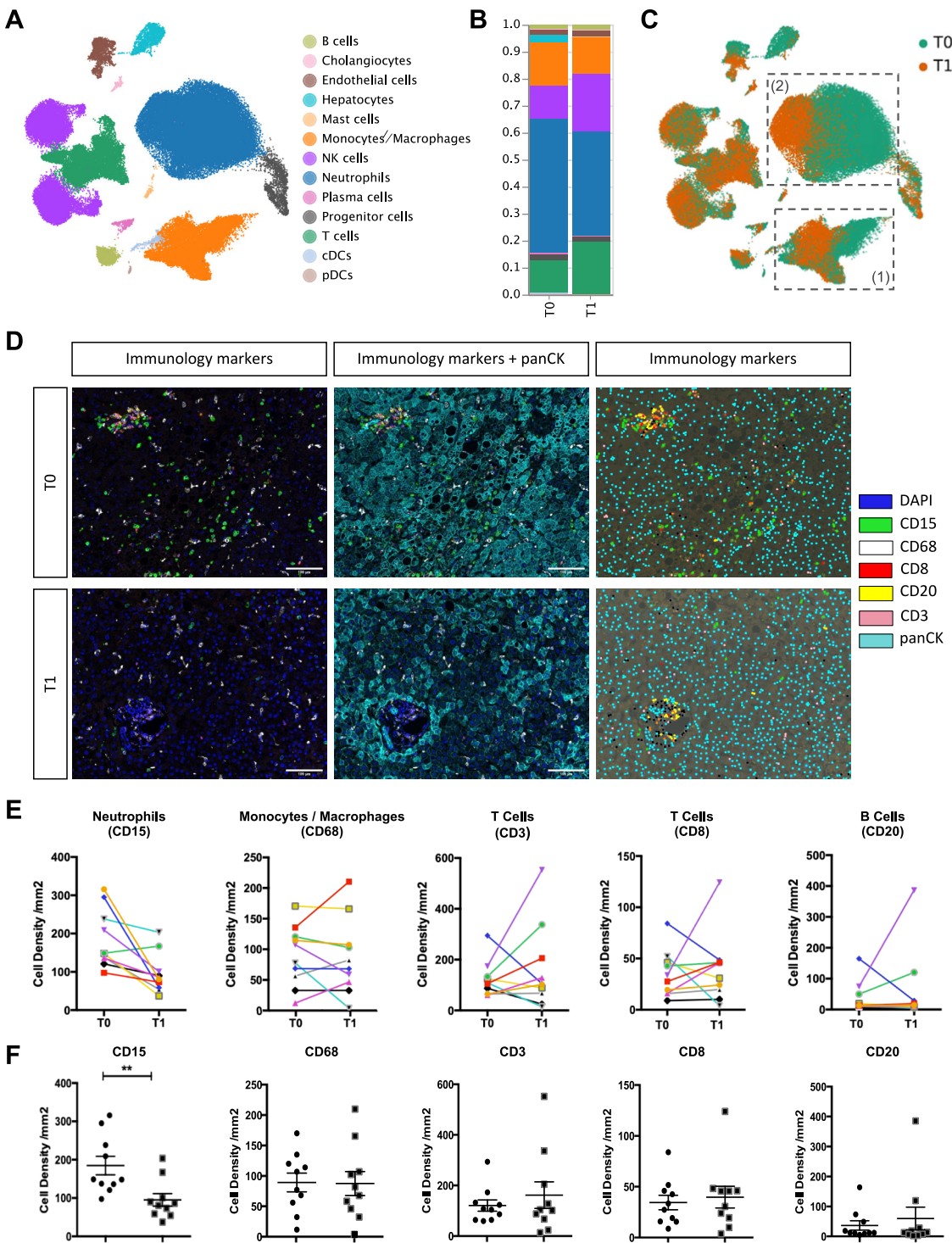

**Fig. 3 | Impact of NMP on immune cells within the liver. A** UMAP plot of 90,404 single cells (only T0 and T1 samples), color-coded by cell type. **B** Relative cell type composition in liver tissues pre (T0) and at the end of (T1) NMP. **C** UMAP-plot of 90,404 single-cells, color-coded by time-point [pre (T0) and at the end of (T1) NMP]. The monocyte/macrophage (1) and neutrophil (2) clusters are highlighted. **D** Multiplex Immunofluorescence images of immunology markers presented alone and together with pan-Cytokeratin and the phenotyping map pre (T0) and at the end of (T1) NMP. The phenotyping map was generated in InForm, each color dot represents a phenotype of the cell, black dots: other cells of unknown phenotype. Images are displayed at ×20 magnification (scale bar: 100 μm). **E, F** Cell densities (number of cells/mm²) for individual immunology markers in 10 patients. The upper panel **E** presents the values for each individual liver pre (T0) and at the end of (T1) NMP. The lower panel **F** presents a column statistical analysis for each biomarker ($n = 10$, paired $t$-test, two-tailed, mean ± SEM, **$p = 0.0097$).

(Fig. 4C, Supplementary Fig. 6B). Hence, NMP resulted in a significant reduction of the *CXCR2* and *CXCR1* gene expression levels (Fig. 4D). We validated the expression of the corresponding CXC chemokine receptor (CXCR)2 protein on neutrophils by multiplex IF staining of 10 livers pre and at the end of NMP (Fig. 4E). Similar to the decreasing neutrophil numbers (Fig. 3D–F), CXCR2 protein expression on neutrophils was significantly reduced during NMP (Fig. 4F). Im parallel, *CXCR4* mRNA expression was markedly elevated during NMP,

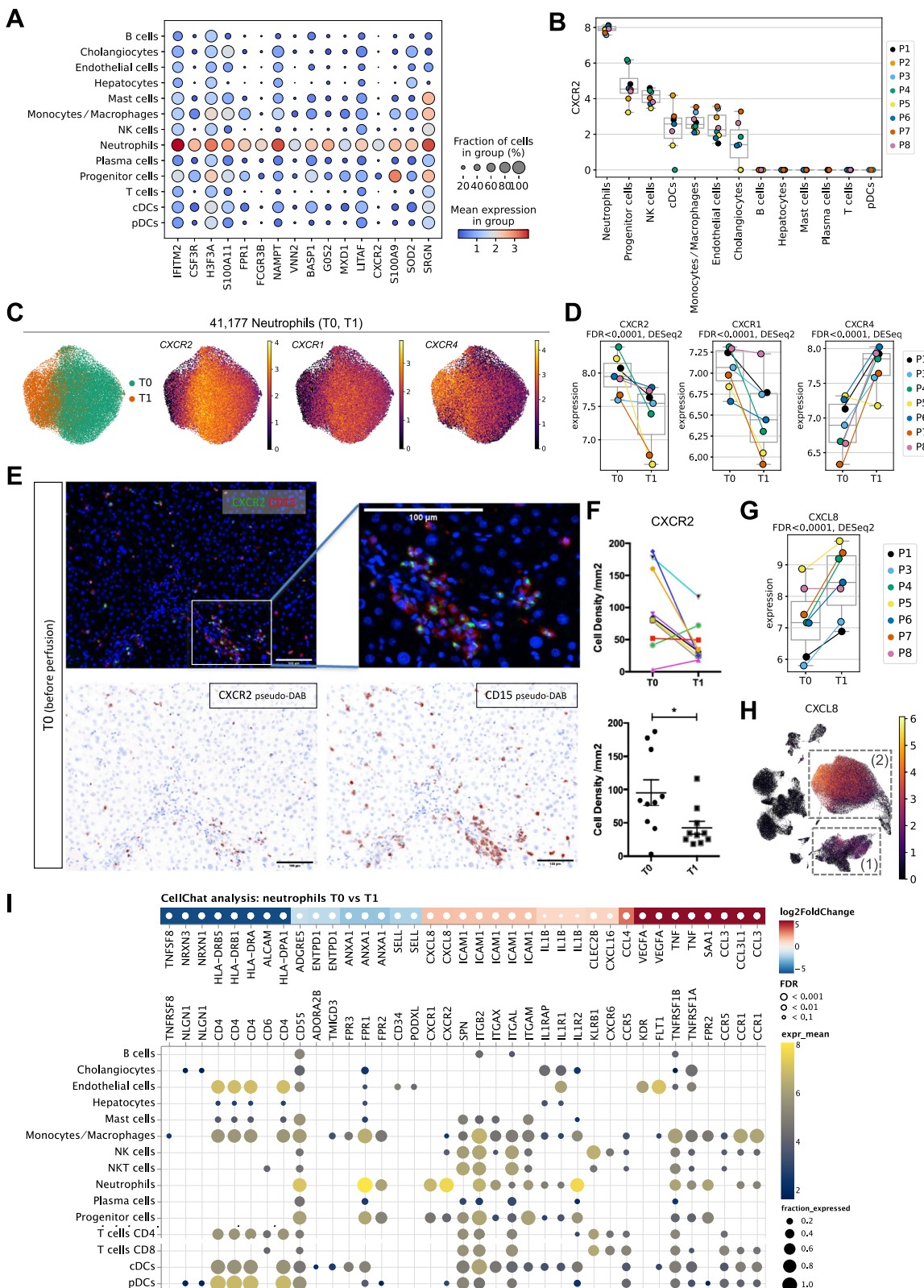

indicating a potential increase in the proportion of exhausted/aged neutrophils (Fig. 4D)[27].

The impact of NMP on the gene signature of neutrophils was further assessed through differentially expressed gene (DEG) analysis (T0 vs. T1; top regulated and selected DEG are shown in Supplementary Fig. 6C). We identified significant up-regulation of *CXCL8*/*IL-8* in the neutrophil cluster during NMP (Fig. 4G; Source Data file). CXCL8 is a cognate pro-inflammatory ligand for both, CXCR2 and

CXCR1 and which mediates neutrophil activation and neutrophil extracellular trap (NET) formation[28]. Neutrophils and to a lower extent monocytes/macrophages were the prime sources for *CXCL8* production (Fig. 4H). Concordantly, analysis of cell-to-cell communication networks using CellChat[29], indicated autocrine neutrophil activation (Fig. 4I) as well as monocyte/macrophage-triggered neutrophil activation (see below) via *CXCL8-CXCR1/CXCR2* signaling during NMP, respectively.

**Fig. 4 | Impact of NMP on neutrophils. A** Gene expression levels of neutrophil-specific genes in individual cell types. **B** *CXCR2* gene expression levels in individual cell types. Each dot represents the mean of a patient (*n* = 8). **C** UMAP plots of 41,177 neutrophils, color-coded by timepoint and the *CXCR2, CXCR1,* and *CXCR4* gene expression levels. **D** *CXCR2, CXCR1,* and *CXCR4* gene expression levels in neutrophils pre (T0) and at the end of (T1) NMP. Each dot represents the mean of a patient (*n* = 7). False-discovery rates (FDR) have been determined using pseudo-bulk analysis with DESeq2. **E** Immunofluorescent staining of CXCR2 and CD15 pre (T0) NMP. Images are displayed at ×20 magnification (scale bar: 100 μm, *n* = 10, for each sample 4 representative images were taken). **F** Cell density (number of cells/mm2) of CXCR2+ cells pre (T0) and at the end of (T1) NMP. The upper graph presents the values for each individual liver, the lower graph is a column statistical analysis (*n* = 10, paired *t*-test, two-tailed, mean ± SEM, *\*p* = 0.02). **G** *CXCL8* gene expression level in neutrophils pre (T0) and at the end of (T1) NMP. Each dot represents the mean of a patient (*n* = 7). False-discovery rates (FDR) have been determined using pseudo-bulk analysis with DESeq2. **H** UMAP plot of 90,404 single cells (only T0 and T1 samples), colored by *CXCL8* gene expression. The monocyts/macrophage (1) and neutrophil (2) clusters are highlighted. **I** Differential signaling from neutrophils to other cell types. Upper panel: Differentially expressed ligands of neutrophils in T0 vs. T1 (two-sided DESeq2 Wald-test on pseudo-bulk, *p*-values were adjusted to false-discovery rate (FDR)). Red colors indicate upregulation in T1 compared to T0. Lower panel: Respective receptors and the expression by cell type. Dot sizes and colors refer to the fraction of cells expressing the receptor and gene expression, respectively, averaged over all patients. Dots are only shown for receptors that are expressed in at least 10% of the respective cell types. In all boxplots, the central line denotes the median. Boxes represent the interquartile range (IQR) of the data, whiskers extend to the most extreme data points within 1.5 times the IQR.

The transcriptional profile of neutrophils at the end of NMP resembled a phenotype that we recently identified and characterized as aged/chronically activated/exhausted neutrophils in non-small cell lung cancer tumor tissues (Fig. 5A: reduced expression of *CXCR1, CXCR2, PTGS2, SELL,* and elevated expression of *CXCR4, CD83, CCRL2, CCL3, CCL4, ICAM1*)[26]. *VEGFA* up-regulation (Fig. 5A, Supplementary Fig. 6C) as well as elevated *VEGFA-KDR1/FLT1* cell communication patterns from neutrophils towards endothelial cells (Fig. 4I), suggests that during NMP tissue repair is induced via re-vascularization of damaged tissue.

Next, we separated neutrophils by unsupervised Leiden-clustering into four subsets (N0–N3; Fig. 5B; top regulated genes are shown in Source Data file), which was backed by all patients (Supplementary Data 6D). The N0 cluster was characterized by the expression of pro-inflammatory genes inducing neutrophil migration to sites of inflammation (*S100A12, S100A8, S100A9, MMP8, MMP9*)[30–32], the NETosis co-factor *PADI4*[33], as well as genes involved in integrin-mediated cell adhesion (*ITGAM, ITGA1, ITGA6*). Hence, the N0 cluster represents a pro-inflammatory, highly activated neutrophil phenotype. The N1 cluster was rather similar to the N0 cluster but showed elevated expression of *ITGA4* as well as of *TAFA4* which modulates macrophage tissue repair functions[34] (Fig. 5C, Source Data file). Notably, the proportion of the pro-inflammatory N0 and N1 neutrophils clearly decreased over the course of NMP, whereas N2 neutrophils were markedly enriched (Fig. 5D). The N2 cluster resembled the above described aged/chronically activated/exhausted phenotype (*CXCR4, CD83, CCRL2, CCL3, CCL4, ICAM1, VEGFA*). N2 neutrophils also expressed *OLR1* (Fig. 5C, Source Data file), a marker we recently identified in chronically activated tumor-associated neutrophils[26]. Concordantly, transcription factor (TF) activity analysis revealed that NMP triggered a significant induction of *PPARG* (Fig. 5E), a direct transcriptional regulator of *OLR1*[35]. The N3 cluster showed high expression of mitochondrial genes (*MT-CYB, MT-ND4, MT-ATP6*), indicating the onset of cellular apoptosis, as well as genes involved in acute phase response (*SAA1, SAA2*) and interferon signaling (*IFIT1, IFIT2*)[36] (Fig. 5C, Source Data file). Notably, N3 neutrophils decreased during NMP (Fig. 5D).

Taken together our findings indicate that NMP shifts neutrophils from an activated pro-inflammatory state towards an aged/chronically activated/exhausted phenotype in human livers, suggesting attenuation of acute inflammatory processes triggered by neutrophils during NMP.

### NMP alters inflammation and tissue repair pathways in human donor livers

In order to assess associations between expression profiles of distinct cell populations and selected gene sets indicative of cellular processes and pathways, gene set enrichment analyses (GSEA) were performed (Fig. 5F). Several pathways involved in inflammatory responses, apoptosis and tumor suppressor activities were recognized in neutrophils. Monocytes/macrophages signalized pathways implicated in complement effector functions. Complement activation is involved in chronic inflammatory processes, but also supports an immunosuppressive microenvironment and induces angiogenesis as well as tissue repair[37]. These findings suggest a shift towards monocytes/macrophages with anti-inflammatory, tissue-healing and regeneration properties.

### During NMP anti-inflammatory/tolerogenic monocytes/macrophages are expanded

Since monocytes/macrophages have a key regulatory function in hepatic inflammatory responses, we assessed their phenotype in greater detail. Monocytes/macrophages characterized by *CD68* cell surface marker expression were significantly altered in their transcriptomic profile during NMP (Figs. 3C, 6A, Supplementary Fig. 7A; Fig. 6B: known marker genes in monocytes/macrophages cells *CST3, CTSB, MS4A7, MARCH1, CD68, MAFB, CD163, VCAN,* and *CSF1R*)[38,39]. The elevated expression of *CXCL8* in monocytes/macrophages (Fig. 4H) and the significant induction of pro-inflammatory chemokines *CCL2* and *CCL3* (selected top DEG in monocytes/macrophages cells pre and at the end of NMP are shown in Supplementary Fig. 7C) indicate that during NMP, the phenotype of monocytes/macrophages shifts towards a pro-inflammatory state. Cell-2-cell communication analysis indicated strongly elevated SPP1 (osteopontin) signaling towards its cognate receptors (e.g. *CD44*) expressed on a variety of immune cells (Fig. 6G). Ostepontin affects acute and chronic inflammation as it regulates immune cell migration, polarization, and activation[40]. Contrary to pro-inflammatory stimulation, NMP decreased the expression of key genes characterizing the inflammatory state of monocyes/macrophages (*LYZ, FCN1, VCAN, HLA-DRA, S100A8, S100A9, S100A12, MNDA, CSTA, CD74*) and elevated the expression level of markers associated with a tolerogenic phenotype (*CD163, MARCO, HMOX1, VSIG4, NSMAF, CTSB, VMO1*; Fig. 6B and Supplementary Fig. 7C)[17]. Traditionally, macrophages are classified either as inflammatory or immune-regulatory[41]. *VSIG4* gene expression mediates T and natural killer T (NKT) cell tolerance during immune-mediated liver injury[42]. Similarly, *HMOX1* (hemoxygenase) knockdown in mice leads to hepatic inflammation[43]. MacParland et al. demonstrated that the expression of *MARCO* is predominantly elevated in non-inflammatory human macrophages[17].

To further dissect the plasticity of monocytes/macrophages during NMP we performed unsupervised Leiden-clustering and identified four distinctive subsets (M0 to M3) (Fig. 6D, Fig. 6E; top regulated marker genes are shown in Source Data file) in all livers (Supplementary Fig. 7D). The M0 subcluster showed high expression of several pro-inflammatory genes, suggesting that this cluster represents inflammatory monocytes/macrophages (*LYZ, VCAN, S100A8, S100A9, S100A12, MNDA*). In line with the DEG analysis, the proportion of pro-inflammatory M0 monocytes/macrophages was strongly reduced during NMP (Fig. 6F). The M1 subcluster was strikingly similar to a frequently reported C1Q+ macrophage population (*C1QB, MRC1, HLA-DOA, FOLR2*)[44]. The observed

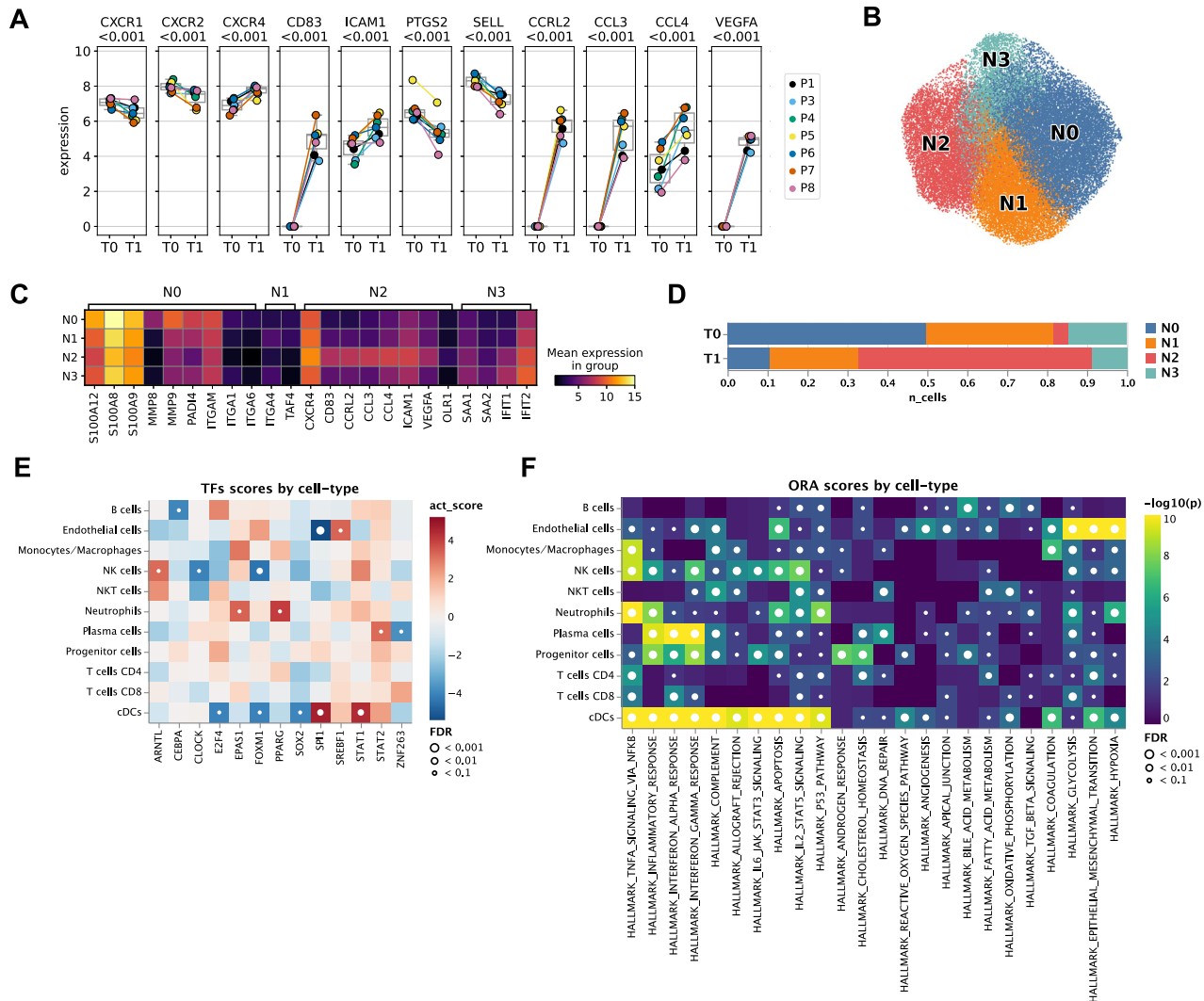

**Fig. 5 | NMP shifts neutrophils towards an aged/chronically activated/ exhausted phenotype. A** Expression of selected marker genes in neutrophils pre (T0) and at the end of (T1) NMP in individual patients ($n = 7$). The central line denotes the median. Boxes represent the interquartile range (IQR) of the data, whiskers extend to the most extreme data points within 1.5 times the IQR. Two-sided DESeq2 Wald-test on pseudo-bulk, $p$-values are adjusted to false-discovery rate (FDR) using independent hypothesis weighting (IHW). **B** UMAP of neutrophils colored by subclusters (N0, N1, N2, N3). **C** Expression of selected marker genes in neutrophil subclusters. **D** Relative composition of neutrophil subclusters pre (T0)

and at the end of (T1) NMP. **E** Differential transcription factor activity (TF) activity per cell-type in T0 vs. T1 computed using DoRothEA. Red indicates upregulation of a TF-regulon in T1 compared to T0. $p$-values were determined using a two-sided $t$-test and adjusted for false-discovery rate (FDR). The $t$-statistics were computed using a multivariate linear model as implemented in decoupler-py. **F** Gene set over-representation analysis (ORA) of selected "Hallmark" gene sets per cell type in T0 vs. T1. A small $p$-value indicates that among genes in the gene set, more genes are differentially expressed than expected by chance (one-tailed Fisher's exact test).

upregulation of complement pathway-related genes (*C1QB, C1QC, AXL*) suggest a potential role of the M1 subcluster in the clearance of injured tissue via efferocytosis[45,46]. On the contrary, the M2 cluster showed a gene expressional profile comparable to non-classical monocytes (*CDKN1C, CYFIP2*)[47,48], as well as high expression of *CX3CR1*, a marker that defines monocytes homing constitutively to tissues and contribute to tissue repair and wound healing[49]. While the M1 and M2 subclusters were not markedly changed during NMP, the M3 cluster was significantly increased (Fig. 6F). This subcluster was characterized by the expression of markers predominantly associated with alternatively-activated ("M2-like") macrophages. M2-like macrophages induce cell proliferation and angiogenesis but are also involved in clearing cell debris and fostering tissue repair (*PLAU, CXCL5, CFS1, FPR3,* SPP*1, CTSL, IL4I1, HS3ST1, SER-PINB2, SLC7A11*)[50–61]. The M3 subcluster also expressed *FN1* (fibronectin), which is involved in extracellular matrix formation and wound repair mechanisms[62]. Of note, cell-2-cell communication analysis in mono-cytes/macrophages indicated elevated *FN1* signaling, e.g. towards its

cognate receptor *ITGB1* expressed on cholangiocytes (Fig. 6G). Additionally, numerous anti-inflammatory markers were identified in the M3 cluster: the *PHLDA1, MMP19, FABP5, NRP1, NRP2, RASGRP3, DHRS9, MT1H*, markers indicate a potential attenuation of pro-inflammatory cytokine production[63–71].

In summary, our data showed that the induction of pro-inflammatory makers and an overall pro-inflammatory phenotype of monocytes/macrophages is accompanied by the induction of anti-inflammatory and tolerogenic ("M2-like") cell subtypes contributing to cell proliferation, angiogenesis, wound healing and tissue repair.

## Rapid migration and washout of leukocytes, primarily neutrophil granulocytes, into the perfusate during liver NMP

After defining the single-cell landscape of tissue-resident immune cells during NMP, we next focused on the perfusate compartment. Therefore, we investigated immune cell dynamics in the perfusate of an additional 26 livers and linked these observations to intra-hepatic

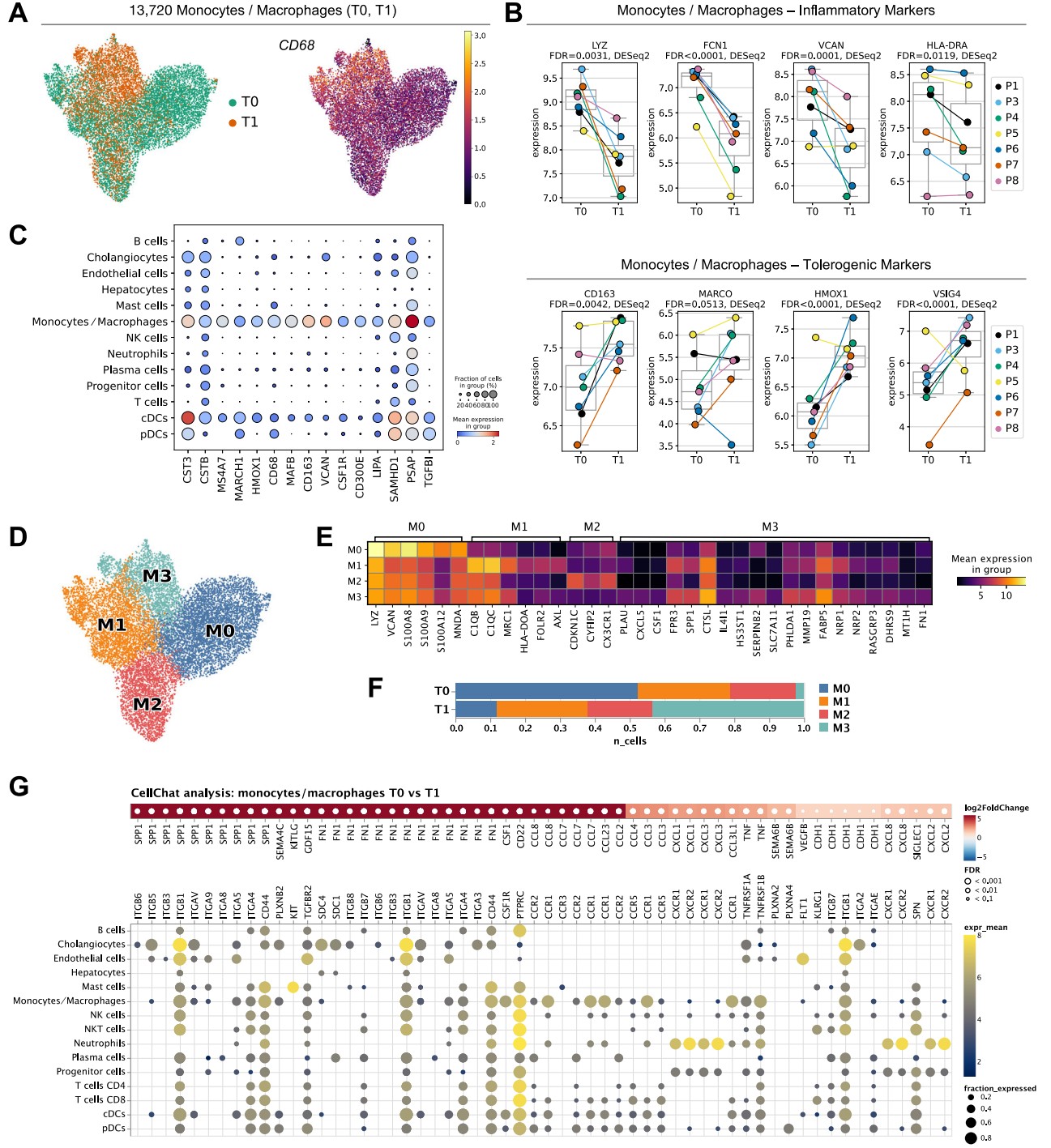

**Fig. 6 | Impact of NMP on monocytes/macrophages. A** UMAP-plots of 13,720 monocytes/macrophages, color-coded by time-point [pre (T0) and at the end of (T1) NMP], and of the relative *CD68* gene expression level. **B** Gene expression levels of inflammatory markers (*LYZ, FCN1, VCAN, HLA-DRA) and* tolerogenic markers (*CD163, MARCO, HMOX1*, and *VSIG4*) in monocytes/macrophages pre (T0) and at the end of NMP (T1). Each dot represents the mean of a patient (*n* = 7). False-discovery rates (FDR) have been determined using pseudo-bulk analysis with DESeq2. The central line denotes the median. Boxes represent the interquartile range (IQR) of the data, whiskers extend to the most extreme data points within 1.5 times the IQR. **C** Gene expression levels of monocyte/macrophage-specific genes. **D** UMAP of monocytes/macrophages colored by subclusters (M0, M1, M2, M3). **E** Expression of selected marker genes in monocyte/macrophage subclusters.

**F** Relative composition of monocyte/macrophage subclusters pre (T0) and at the end of (T1) NMP. **G** Differential signaling from monocytes/macrophages to other cell-types. Upper panel: Differentially expressed ligands of monocytes/macrophages in T0 vs. T1 (two-sided DESeq2 Wald-test on pseudo-bulk, *p*-values were adjusted to false-discovery rate (FDR)). Red colors indicate upregulation in T1 compared to T0. Lower panel: Respective receptors and the expression by cell type. Dot sizes and colors refer to the fraction of cells expressing the receptor and gene expression, respectively, averaged over all patients. Dots are only shown for receptors that are expressed in at least 10% of the respective cell types. Only upregulated ligands are shown. The list of downregulated interactions is available in the Source Data file.

immune cell alterations during NMP (Fig. 7A). Cell viability was confirmed in a series of perfusate samples (Fig. 7B). As early as 1 h after NMP initiation, a rapid increase of absolute CD45[+] leukocyte numbers was observed (2537/μl [2003, 3215]). The dominating subsets were CD45[+]HLA-DR[low] granulocytes (1947/μl [1510, 2511]), CD45[+]CD3[+] T cells (203/μl [157, 263]), CD45[+]CD56[+] NK cells (99/μl [68, 143]), CD45[+]CD19[+] B cells (52/μl [38, 71]), and CD45[+]CD14[+] monocytes/macrophages (37/μl [26, 54]) (Fig. 7C, D). Regression analysis indicated a statistically significant change in absolute cell numbers over perfusion time for all leukocyte subtypes. While CD45[+]CD19[+] B cells and CD45[+]CD14[+] monocytes/macrophages peaked at 6 h, a decline in the total leukocyte number was observed for all subsets beyond 6 h of NMP (Fig. 7E, F, Supplementary Table 2).

The proportion of neutrophils as identified in the scRNASeq immune cell atlas and IF staining in liver tissue decreased over NMP time, while a high absolute number of neutrophils was found in the perfusate. These corresponding findings indicate a swift migration of neutrophils into the perfusate. This holds also true—albeit to a lesser extent—for other inflammatory cells. These cellular shifts correspond with the patterns of immune cells obtained in liver biopsies with respect to their basic quantitative patterns and dynamics. In summary, a rapid and significant migration primarily of neutrophils into the perfusate was found.

### In-depth phenotyping of immune cell subtypes in serial perfusate samples during NMP

We further characterized subtypes of the main immune cell populations and their dynamic changes in five serial perfusate samples using flow cytometry. At 1 h NMP, CD45[+]CD3[+] T cell subsets in the perfusate comprised 48% [42, 54] CD3[+]CD4[+] T helper cells, while 44% [37, 50] were CD3[+]CD8[+] cytotoxic T cells. The proportion of both subsets significantly changed in favor of CD4[+] T helper cells over perfusion time. CD3[+]CD56[+] NKT cells and CD3[+] T cell receptor (TCR) Vα2.2[+]CD161[+] mucosal-associated invariant T (MAIT) cells represented only small proportions of total T cells without relevant dynamics over perfusion time (Fig. 7G). Proportions of CD3[+]CD4[+] memory T cells significantly changed over perfusion time (Fig. 8A), whereas CD3[+]CD8[+] memory T cells remained stable (Fig. 8B). CD4[+]CD25[+]FoxP3[+] Tregs comprised 2.32% [1.58, 3.28] of the T cell population at 1 h. Notably, their proportion significantly increased over perfusion time to 4.71% [3.29, 6.59] at 24 h NMP (Fig. 8C). CD45[+]CD56[+] NK cells were mostly CD56[dim]CD16[+] (94.21% [96.11, 91.58]) without significant changes over time (Fig. 8D). Amongst the CD14[+]CD16[+] monocytes/macrophages, CD14[+]CD64[+]CD163[+] Kupffer cells comprised only small proportions in the perfusate at 1 h (5.56% [3.39, 8.78]), but a strong proportional increase with prolonged perfusion was observed (24 h NMP: 11.86% [7.05, 19.52], Fig. 8E). Among numerically prominent subtypes (i.e. CD45[+]HLA-DR[low] granulocytes), CD15[+]CD16[+] neutrophils were dominating (93.45%% [88.90, 96.30]) over very few Siglec[-]CD8[-]CD16[-] eosinophils (0.38% [0.20, 0.59]), and CD16[+]CD123[+] basophils (0.23% [0.13, 0.34]) at 1 h NMP. Their proportions remained unchanged over time (Fig. 8F). Dendritic cells represented only a small proportion of total leukocytes (0.86% [0.53, 1.27] at 1 h NMP) without significant dynamics during NMP (Fig. 8G, Supplementary Table 3).

These data highlight the mobilization of a broad leukocytes spectrum into the circulating perfusate during NMP. The analysis of serial samples taken during the course of the 24 h NMP period revealed distinct migration dynamics. While migration of a number of immune cell subtypes into the perfusate appeared very quickly, mobilization and release of Tregs and Kupffer cells increased as early as 12 h after NMP.

### Monocytes/macrophages are the main source of interleukins/chemokines released into the perfusate during NMP

We next investigated interleukin/chemokine gene expression levels in immune cells and protein levels in serial perfusate samples. While monocytes/macrophages increasingly expressed pro-inflammatory *IL1B*, *IL18*, *CXCL8*, *CCL2*, *CCL3*, *CCL4*, and *TNF*, neutrophils mainly expressed *CXCL8* and – to a lesser extend – *IL1B* and *CXCL1* (Fig. 8H). The expression of anti-inflammatory markers such as *IL10* was induced in monocytes/macrophages (Fig. 8H)[72].

In line with the transcriptomic dataset, pro-inflammatory cytokines expressed in monocytes/macrophages and neutrophils also increased in the perfusate over time, denoting increased transcription and protein release into the hepatic microenvironment and the perfusate (Fig. 8I). Supplementary Figure 8 depicts the dynamics of all cytokines/chemokines on protein level over perfusion time (the respective values are given in Supplementary Table 4). Only mild alterations were found in the transcriptomic signatures of T and NK cells (Supplementary Fig. 9).

### High perfusate interleukin (IL)-6 levels are attributed to monocytes/macrophages and linked to donation after cardiac death (DCD) grafts and discarded livers

The impact of donor type (donation after brain death (DBD) vs DCD) and suitability for transplantation (transplanted vs discarded livers) on protein cytokine dynamics was assessed in more detail. Most perfusate cytokines were increased with prolonged perfusion time in all groups. IL-6 concentrations were most significantly elevated in perfusate of DCD grafts as compared to DBD grafts ($p = 0.05$, Supplementary Fig. 10, Supplementary Table 5). This phenomenon was also observed when comparing discarded livers vs. transplanted livers ($p = 0.032$, Fig. 8J, Supplementary Fig. 10, Supplementary Table 6). Concordantly we found elevated *IL6* expression in monocytes/macrophages at the transcriptional level, indicating that this cell type at least substantially contributes to IL-6 production (Fig. 8J). Further to IL-6, also tumor necrosis factor (TNF) was significantly increased in the perfusate when comparing discarded vs transplanted livers ($p = 0.012$, Fig. 8J, Supplementary Fig. 11, Supplementary Table 6).

Taken together our data suggest that the immunologic response during liver NMP varies between donor types and "low quality" (not suitable for transplantation) vs. "acceptable" grafts.

## Discussion

Ex vivo NMP of human organs is a rapidly advancing clinical tool to improve and prolong organ preservation. Further, the ability to assess the function prior to transplantation addresses an unmet need and offers a unique possibility to elucidate the molecular events during reperfusion. This technology may also set the stage for future targeted ex vivo organ treatment. As the liver contains a huge amount of immune cells with important immune-regulatory functions and with central importance for adequate liver function[73], we in-depth characterized immune cell (sub)populations in human donor livers and perfusate during NMP and assessed perfusate cytokines. Deep immune cell mapping indicated a predominance of neutrophils in the donor's livers, which are promptly mobilized into the perfusate during NMP. Neutrophils transit from an activated pro-inflammatory state towards an aged/chronically activated/exhausted phenotype and monocytes/macrophages express anti-inflammatory/tolerogenic characteristics essential for tissue repair.

NMP in 34 human donor livers included in this study was uneventful and allowed for repeat assessment of tissue and perfusate samples without impact on organ integrity. Neutrophils were identified as the most prevalent immune cell population-based sc-transcriptomics performed on eight donor livers. Multiplex IF and IHC analysis corroborated that CD15[+] neutrophils are the dominating hepatic immune cell type, distributed over the whole organ in a spotted pattern. Intriguingly, to the best of our knowledge, the neutrophil lineage is entirely dismissed in hitherto published scRNASeq datasets of human livers[17–19,22–24,74]. This discrepancy is most likely attributed to methodological pitfalls rather than to a biological

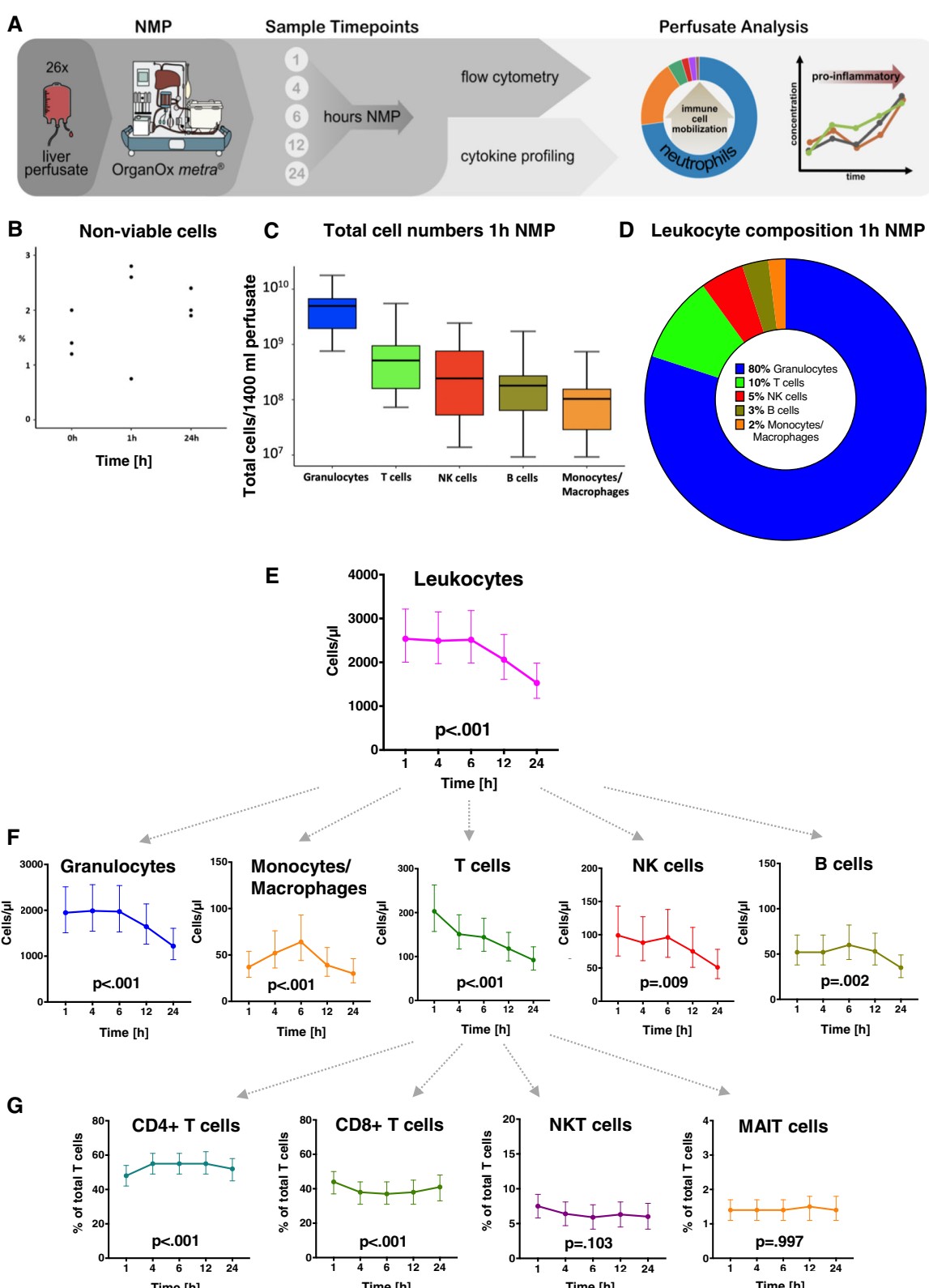

**Fig. 7 | Total release of passenger leukocytes into the perfusate (*n* = 26) at various time points during NMP. A** NMP perfusate analysis workflow and applied methods. **B** Cell viability testing prior to flow cytometry (*n* = 3 pre, at 1 h NMP and at the end of NMP) showed only a very small percentage of non-viable cells in the perfusate. **C** Absolute CD45⁺ leukocyte amounts for main immune cell subtypes in total circulating perfusate (mean ± SEM) and **D** composition at 1 h NMP. **E**, **F** Dynamic change of total CD45⁺ leukocytes and main subtypes during NMP.

**G** Dynamic change of CD3⁺ T cell subtypes (proportions) over perfusion time. Graphs show the marginal effects. The values are estimated using linear regression analysis. The *p*-values refer to the change over time. The least-squares means computed using a linear model are shown together with the 95% CI. *N* = 26 biologically independent samples. Source data are provided as a Source Data file. NK cells natural killer cells, NKT cells natural killer T cells, MAIT cells mucosa-associated invariant T cells.

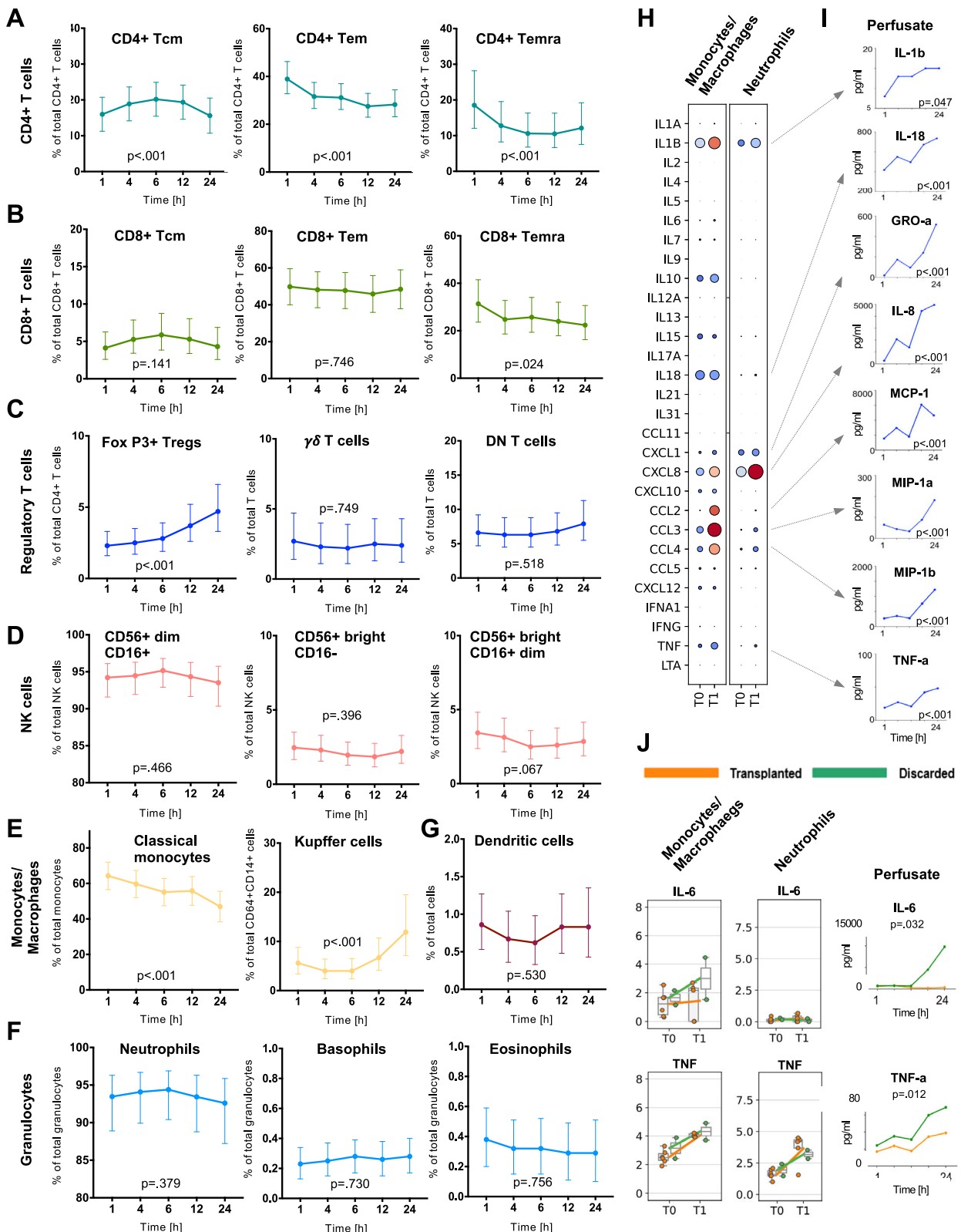

phenomenon. Because neutrophils are a remarkably short-lived[75] and extremely fragile cell type that is particularly sensitive to tissue dissociation and sorting, a quick and yet gentle workflow from tissue dissociation to cell lysis is essential to preserve these cells. Moreover, neutrophils express an exceptionally low amount of mRNA molecules[76], which impedes their recovery in scRNASeq data. We recently demonstrated that neutrophils cannot be appropriately

detected in datasets generated with the droplet-based 10x Chromium sequencing platform and only to a very limited extent with other platforms. However, the microwell-based scRNASeq platform applied in this study allows to capture of relatively high numbers of mRNA molecules per cell and thus enhances the recovery of low mRNA content cells[26]. While scRNASeq of eight livers generated a huge dataset, the total number and the spectrum of differences in liver

**Fig. 8 | Phenotyping of perfusate immune cells and inflammatory profile during NMP.** Perfusates collected at various time points during NMP ($n = 26$ livers subjected to NMP) were assessed for their dynamic changes of the proportions of **A** CD4$^+$ T cells, **B** CD8$^+$ T cells, **C** subtypes of CD3$^+$ T cells with regulatory properties, **D** CD56$^+$ NK cells, **E** CD45$^+$CD14$^+$ monocytes/macrophages, **F** CD45$^+$HLA-DR$^{low}$ granulocytes, and **G** dendritic cells. Graphs show the marginal effects. The values are estimated using linear regression analysis. The p-values refer to the change over time. The Least-Squares Means computed using a linear model is shown together with the 95% CI. $N = 26$ biologically independent samples. **H** Gene expression levels of indicated pro-inflammatory factors (interleukins/chemokines) in monocytes/macrophages and neutrophils pre (T0) and at the end of NMP (T1) as assessed by scRNASeq in eight donor livers. **I** Spectrum of assessed pro-inflammatory interleukins/chemokines produced by monocytes/macrophages and neutrophils and elevated on protein level in perfusate samples collected over perfusion time of 26

donor livers. (J) Gene expression levels of *IL-6* and *TNF* in monocytes/macrophages and neutrophils pre (T0) and at the end of NMP (T1) as assessed by scRNASeq in transplanted ($n = 6$) and discarded ($n = 2$) livers. The central line denotes the median. Boxes represent the interquartile range (IQR) of the data, whiskers extend to the most extreme data points within 1.5 times the IQR. In addition, the perfusate of 26 donor livers subjected to NMP was measured for IL-6 and TNF levels at 1, 4, 6, 12 and 24 h of NMP and differences between transplanted ($n = 18$) and discarded ($n = 8$) livers were calculated. Graphs show the marginal effects. The values are estimated using linear regression analysis. The *p*-values refer to the change over time. The Least Squares Means computed using a linear model are shown together with the 95% CI. $N = 26$ biologically independent samples. Source data are provided as a Source Data file. Tcm: central memory T cells, Tem: effector memory T cells, Temra: effector memory cells re-expressing CD45RA T cells, DN T cells: double-negative T cells (CD4$^-$ and CD8$^-$), NK cells Natural killer cells, DCs dendritic cells.

quality remain a limitation. Even though we included ECD livers in this study, the organs may vary in graft quality and pre-existing disease/damage, which makes it impossible to capture the full breadth of inter-individual variation in this effort.

Neutrophil-driven inflammatory circuits, as observed in our NMP perfusates, have been shown to promote various acute and chronic liver pathologies[77]. Similar to neutrophils, monocytes/macrophages are key regulators of tissue homeostasis and immune activation within the liver[78]. We here provide evidence that NMP triggers a shift from highly activated neutrophils towards an aged/chronically activated/exhausted phenotype in human livers. Specifically, neutrophil activation by NMP is mirrored by induction of the IL-8/CXCR1/2 axis, which contributes to the pro-inflammatory micromilieu. This may also lead to induced NET formation resulting in aggravated hepatic IRI[79]. These findings suggest that CXCR1/2 inhibition during neutrophil activation may help to limit an excessive inflammatory response during NMP.

The dynamics of monocytes/macrophages during NMP cannot be unambiguously described as a pro- or anti-inflammatory phenotype but rather indicates a response and counter-response sequence. This is also reflected by subcluster analysis demonstrating that pro-inflammatory monocytes/macrophages are diminished, while anti-inflammatory and tolerogenic ("M2-like") subtypes involved in cell proliferation, angiogenesis, wound healing and tissue repair are significantly increased during NMP, respectively.

The current understanding of the impact of NMP (and machine perfusion in general) on the inflammatory profile/immune cell repertoire of an organ is limited[80]. In rat livers, damage-associated molecular patterns (DAMPs) were shown to induce tissue injury during machine perfusion[81]. In a similar model, NMP was shown to induce a broad inflammatory gene expression signature and activation of liver-resident immune cells. Treatment with IL-10 and transforming growth factor (TGF)-ß resulted in moderation of inflammation in this model[82]. In a study conducted on human livers, NMP inhibited inflammation and promoted graft regeneration[83].

Consistent with enhanced cytokine expression observed in six livers over 6 h NMP by Lee et al., we found both pro-and anti-inflammatory cytokine expression in the perfusate[83]. The accumulation of interleukines/chemokines in the perfusate matches the transcriptomic changes in neutrophils and monocytes/macrophages in the hepatic parenchyma. A significant increase of IL-6 in perfusate of DCD grafts and discarded livers may be indicative of a more pronounced inflammatory response in these organs. Indeed, Ohman et al.[84] recently reported that the immune response in low-quality/inferior-functioning livers differed from functioning grafts when subjected to NMP, although exorbitant high perfusate levels of IL-6 were reported for both groups. Thus, our observations require further attention prior to the determination of their clinical significance.

Leukocyte and in particular neutrophil mobilization and transition into the perfusate was extremely rapid. This is consistent with data

from porcine lungs and kidneys during NMP. In these studies, excessive migration was also linked to reduced immunogenicity with lower acute rejection rates[13,14]. We herein describe perfusate immune cells in great detail and provide insights into dynamic changes for up to 24 h perfusion in a cohort of 26 human livers. While a significant number of cells was present in the perfusate at the end of NMP in our study, the decrease of most cell types with perfusion time seems to indicate mitigation of this effect. Alternatively, circulating immune cells may migrate back into the liver. More detailed and future tracking studies of immune cells between tissue and perfusate will enlighten this dynamic process in more detail.

Moreover, modulation of this process, e.g. by active leukocyte mobilization and/or elimination of leukocytes may reduce the antigenic/inflammatory load of the liver during NMP. However, indiscriminate withdrawal may not be ideal, since subsets of circulating immune cells are also involved in tissue regeneration, tissue remodeling, healing and tolerance induction[80]. As an example, we demonstrate strong induction of an anti-inflammatory "M2-like" phenotype in monocytes/macrophages during NMP. This may foster tissue repair and remodeling mechanisms and we, therefore, suggest a more targeted elimination of inflammatory and potentially tissue-damaging immune cell subsets. Here, we suggest that selective targeting of pro-inflammatory neutrophils (e.g. by CXCR2 antagonists) could be a first step towards targeted immunomodulation during ex vivo perfusion. Moreover, filtration of pro-inflammatory cytokines during NMP might also be beneficial. First results from a human kidney perfusion trial indicated a positive effect on delayed graft function-associated gene expression signatures[85].

In conclusion, our comprehensive analysis highlights the dynamics of the hepatic immune cell repertoire during NMP of human livers on a transcriptomic and protein level. In-depth immune cell mapping revealed the predominance of neutrophils in human donor livers. Together with a variety of other immune cell populations, these cells rapidly transmigrate at a great extent into the perfusate during liver NMP. With the prolongation of perfusion, the immune cell activation status diverges from a pro-inflammatory to an exhausted but also regenerative phenotype. Our findings and the corresponding datasets may serve as a basis for future interventional studies and open further avenues to mitigate inflammation by targeted immunomodulation during NMP and LT.

## Methods

### Study design and normothermic machine perfusion

This research complies with all relevant ethical regulations. The study was approved by the institutional review board (protocol #1175/2018) of the Medical University of Innsbruck. Informed consent from all participants was previously obtained.

Between April 2019 and July 2021, a total of 34 donor livers subjected to NMP were enrolled. All organs were accepted with the intent

to transplant after NMP. Livers were flushed during retrieval using the University of Wisconsin (UW, $n = 14$) or Histidine-tryptophan-ketoglutarate (HTK, $n = 20$) solution and shipped as per standard cold storage conditions using the same preservation fluid on ice. After transportation to our institution (back-to-base concept), livers were perfused using the OrganOx metra device according to local protocols[2]. Upon arrival at our center, grafts were flushed with 2000 ml HTK to remove any remaining blood cells. Livers were then surgically prepared for NMP and connected to the perfusion machine. The NMP perfusate consisted of three units of type O leukocyte-depleted (30 Gy irradiation) packed red blood cells (RBC, 300 ml each), 500 ml Gelofusine (B. Braun) and additives as per the manufacturer's protocol. According to local standards, one packed unit of leukocyte-depleted red cell bag contains a maximum of $1 \times 10^6$ leukocytes. A standardized protocol for NMP including a scheme for perfusate analyses as recently established at our institution was applied[2]. This includes a multidisciplinary approach with organ observation and management at the ICU. The decision to transplant or discard an organ was based on lactate decrease, pH maintenance, and glucose consumption. Maintenance of physiological pH values (7.3–7.45) without sodium bicarbonate supplementation after 2 h NMP as well as a rapid decrease and maintenance of lactate to physiological levels (≤18 mg/dl) are considered key factors indicating good organ function. Further to this, exceptionally high AST, ALT, and lactate dehydrogenase levels (>10,000 U/l) and a sharp incline of these parameters are considered warning signals. Liver recipients included in the study were adults ≥18 years of age, listed for a first or a re-transplantation.

Of 34 donor livers subjected to NMP, scRNASeq analysis was performed in eight randomly selected study livers. Therefore, wedge liver biopsy samples were collected at three individual time-points, i.e. pre-NMP (T0), at end of NMP (T1), and after reperfusion (T2) if transplanted. Serial perfusate samples of another 26 study livers were collected at 1, 4, 6, 12 h and at end of NMP for cell quantification/phenotyping and cytokine assessment by flow cytometry and Luminex analysis. Detailed information on sample numbers, tissue/perfusate collection, time-points and type of analysis are provided in Fig. 1.

### Tissue dissociation for scRNASeq and flow cytometry analysis

Liver biopsies taken pre (T0) and at the end (T1) of NMP as well as after reperfusion (T2) ($n = 8$ livers) were immediately minced into small pieces (<1 mm) on ice and subsequently digested enzymatically for 10 min at 37 °C with agitation using the BD TuDoR dissociation reagent (BD Biosciences). The single-cell suspension was filtered through a 100 μM cell strainer and red blood cells were removed with the BD Pharm Lyse (BD Biosciences) lysing solution according to the manufacturer's protocol. Cell viability was measured with the BD Rhapsody scRNASeq platform (BD Biosciences) using Calcein-AM (Thermo Fisher Scientific) and Draq7 (BD Biosciences).

### scRNASeq library preparation and sequencing

The freshly isolated single-cell suspension was immediately processed (<30 min from tissue dissociation to cell lysis) and whole transcriptome amplification (WTA) sequencing libraries were generated according to the BD Rhapsody single-cell WTA protocol. We selected the microwell-based BD Rhapsody scRNASeq platform with the intention to comprehensively deconvolute leukocyte dynamics, as it allows to depict of low-mRNA-content cells (e.g. neutrophils) and may lead to the loss of large cells >40 μm (hepatocytes) due to a bead exclusion phenomena. The quality of the obtained sequencing libraries was verified with the 4200 TapeStation system (Agilent) and the Qubit dsDNA HS (High Sensitivity) assay kit (Thermo Fisher Scientific). Sequencing was performed on the NovaSeq 6000 System platform (Illumina) with the S1 Reagent Kit v1.5 (200 cycles, 68 bp index read 1; Illumina) at a calculated sequencing depth of 50,000 reads/cell.

### scRNASeq data pre-processing, quality control and analysis

Bioinformatic pre-processing of the obtained FastQ sequencing files was performed via the cloud-based Seven Bridges Platform environment (Seven Bridges Genomics) using the BD Rhapsody WTA Analysis Pipeline app. Data was loaded into AnnData[86] for further processing with scverse tools. Quality control was performed using scanpy[87], only retaining cells with (1) between 250 and 8000 detected genes, (2) between 1000 and 100,000 transcripts, and (3) <30% mitochondrial transcripts. The 4000 most highly variable genes (HVGs) were selected using scanpy's *highly_variable_genes* function with the options *flavor = "seurat_v3"* and *batch_key = "patient"*. Cell transcriptomes were embedded into a batch-corrected low-dimensional latent space using scVI[88,89], treating each sample as a separate batch. Doublets were identified and removed using SOLO[90] as implemented in scvi-tools[89]. A neighborhood graph and uniform manifold approximation and projection (UMAP) embedding[91] were computed based on the scVI latent space. Cell types were annotated based on unsupervised clustering with the Leiden algorithm[92] and known marker genes.

We used DESeq2[93] on pseudo-bulk samples for differential expression testing which has been demonstrated to perform well and properly correct for false discoveries[94]. For each cell type and patient, we summed up transcript counts for each gene that is expressed in at least 5% of cells using decoupler-py[95]. Pseudo-bulk samples consisting of fewer than 1000 counts or 10 cells were discarded. P-values were adjusted for multiple hypothesis testing with independent hypothesis weighting (IHW)[96]. Marker genes for monocyte/macrophage and neutrophil subclusters were determined using the area under the receiver operator characteristics curve (AUROC) and log2 fold-change metrics on pseudo-bulk samples as defined in Becht et al.[91]. Marker genes were defined as having an AUROC > 0.7 and a log2 fold change > 1 for neutrophils and >2 for monocytes/macrophages.

We performed transcription factor analysis with DoROthEA[97] using a multivariate linear model as implemented in decoupler-py[95]. Only regulons with the highest confidence levels "A" and "B" were used. Fold changes from the DESeq2 analysis were used as input. Additionally, we performed gene set enrichment analysis using an over-representation test (ORA, i.e. Fisher's exact test) as implemented in decoupler-py. DE comparisons between different cell-types have different statistical power due to a different number of pseudobulk-samples. To avoid biases due to different numbers of differentially expressed genes per cell type, we used the 182 most differentially expressed for each cell-type for the over-representation test, which corresponds to the median number of DE genes across all cell-types at a false discovery rate (FDR) < 0.01 and |log2 fold change | > 1.

We used the CellChat database[29] as obtained from omnipathdb[29] to investigate differences in cell-to-cell communication between timepoints. The original CellChatDB algorithm is designed to find differences between cell-types. For our study, on the other hand, we were interested in differences between timepoints, using patients as biological replicates. Therefore, for each cell-type of interest, we considered the list of significantly differentially expressed ligands in CellChatDB. For each of those differentially expressed signaling molecules and for each cell-type, we determined interaction partners that are potentially affected by that change, as those that are expressed in at least 10% of the cells in a certain cell-type. Differentially expressed signaling molecules were determined with DESeq2 as described above. The fraction of cells expressing a signaling molecule was computed as the mean of fractions per patient, to avoid biases due to different cell-counts per patient.

### Multiplex immunofluorescence staining

Liver biopsies taken pre (T0) and at the end (T1) of NMP ($n = 10$ randomly selected out of 26 livers) were fixed in 4% paraformaldehyde and embedded in paraffin. Multiplexed IF staining was performed on 4 μm sections using the Opal 7-Color Automated

Immunohistochemistry Kit (cat: NEL821001KT, Akoya Biosciences). A panel of immune markers was configured including antibodies against CD15, CD8, CD68, CD3, CD20, cytokeratin (Panel 1) and CD15, CXCR2 (Panel 2). Markers were sequentially applied and paired with respective Opal fluorophores (antibodies used are listed in Supplementary Table 7). The staining was performed using an automated system (BOND-RX; Leica Biosystems). To visualize cell nuclei, the tissue was stained with 4',6-diamidino-2-phenylindole (spectral DAPI, Akoya Biosciences). Slides were scanned using Mantra 2 Quantitative Pathology Workstation (Akoya Biosciences) and the Mantra Snap software v1.0.4. Five to eight representative images from each tissue were acquired for the analysis. Spectral unmixing, multispectral image analysis, and cell phenotyping were carried out using the inForm Tissue Analysis Software v2.4.10 (Akoya Biosciences). Immune cell density was quantified and is given as "number of cells/mm$^2$". The paired $t$-test was used to assess the differences between the two groups. In grouped analysis, data points are presented on scatter plots with means ± standard error of the mean (SEM). Analyses were performed with GraphPad Prism v6.0.

### Flow cytometry

Cells isolated from liver biopsies pre-NMP (T0) and at the end of NMP (T1) were stained with a cocktail of 19 antibodies (Supplementary Table 8) at pre-titrated concentrations and, after washing and addition of 7-AAD, measured on a FACSymphony A5 flow cytometer. Data were analyzed using FlowJo v10.7 software (for details of the gating strategy see Supplementary Fig. 12).

Serial perfusate samples of 26 livers subjected to NMP (5 ml) were collected in Cyto-Check BCT tubes (Streck) at 1, 4, 6, 12 h and at end of NMP. Immediate cell fixation in these tubes maintains cellular morphology and surface antigen expression and allows storage of samples at room temperature at least for 5 days. In our study fixation tubes were chosen to ease logistics. 500 μl of perfusate was used for every staining. Initially, red blood cells of 500 μl perfusate were lysed with the RBC Lysis Buffer (Thermo Scientific, eBioscience). After blocking with Fc Blocking Reagents (BD Bioscience) cells were incubated with the antibody master mixes in various combinations containing the antibodies referred to in Supplementary Table 9. For the FoxP3 intracellular staining, cells were permeabilized using the Foxp3/Transcription Factor Staining Buffer Set (Thermo Scientific, ebioscience) and incubated with Normal Rat Serum (Thermo Scientific, ebioscience) prior to incubation with the FoxP3 antibody. For quantification of absolute cell numbers, BD Trucount Tubes (BD Bioscience) were used according to the manufacturer´s protocol and analyzed using a LSRFortessa flow cytometer (Becton Dickinson and Company). Doublet exclusion was achieved by plotting forward and sideward scatter areas, heights, and widths (FSC- and SSC-A/-H/-W). Data were analyzed with FlowJo v6.2 (Tree Star). The gating strategy is presented in Supplementary Figs. 13–20. For cell viability testing, the perfusate was collected at NMP start, after 1 h and at the end of NMP ($n = 5$). Red blood cells were lysed with RBC lysis buffer and leukocytes were stained using trypan blue. A minimum of 150 cells/sample were counted in duplicates, results were averaged and the percentage of dead/living cells was calculated.

### Cytokine quantification

Serial perfusate samples of 26 livers subjected to NMP were centrifuged and 500 μl of serum was frozen at −80 °C. Cytokine/chemokine protein levels were measured using the Cytokine&Chemokine 34-Plex Human ProcartaPlex Panel 1A (EPX340-12167-901, Thermo Fisher Scientific) in a Luminex MAGPIX instrument (Luminex Corporation) and analyzed by xPonent 4.2 Rev.2 software (Luminex Corporation) according to the manufacturer´s protocol. Cytokine profiling was read in the light of donor type (DBD vs DCD livers) and transplantation status (transplanted vs discarded livers).

### Statistical analysis

Perfusate data are expressed as absolute values or proportions (%), estimated averages ± standard deviation (SD) and 95% confidence interval (CI). A linear-mixed effect model for repeated measures (LMM) was used for cell composition dynamics. Given the positively skewed distribution, a logarithmic transformation was performed. One-way ANOVA was used to evaluate differences considering the overall time-course perfusion. Single-cell differential gene expression analysis was performed using DESeq2 on pseudo-bulk samples aggregated by biological replicate. Analyses were performed using $R$ statistical software (v4.0.3, Team RC, R Foundation for Statistical Computing)[98]. Graphs were completed using GraphPad Prism 9.1.0 (216). $P$-values for untargeted analyses (DE genes, TFs, gene sets) were FDR-adjusted. Significance levels and more details on the statistical tests are indicated in the figure captions.

### Reporting summary

Further information on research design is available in the Nature Portfolio Reporting Summary linked to this article.

## Data availability

Sequence data that support the findings of this study (all Figures) is available through the NCBI GEO accession GSE216584. All other data are available in the article and its Supplementary files or from the corresponding author upon request. Source data are provided with this paper.

## Code availability

The source code to reproduce the data analysis is available from https://github.com/icbi-lab/nmp-liver. Processed input data and containerized software dependencies required to execute the code are available from zenodo: https://doi.org/10.5281/zenodo.7249006.

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

## Acknowledgements

The authors would like to thank Astrid Drasche and Annabella Pittl for technical support and D.I. Wolfgang Peter for statistical support. The research was supported by the Austrian Science Fund FWF (Grant No. TAI-697) (D.W.), the "In Memoriam Gabriel Salzner Stiftung" (S.S., D.W.), Tiroler Wissenschaftsfond (TH), Jubiläumsfonds— Österreichische Nationalbank (R.O.), FFG grant Austrian Research Promotion Agency, 858057 (HD FACS, S.So.). G.S. was supported by a DOC fellowship of the

Austrian Academy of Sciences. The author Margot Fodor (M.F.) will use part of the materials and data of this study for the completion of her Ph.D. thesis.

## Author contributions

Conceptualization and study design (T.H., R.O., A.P., D.W., S.Sc.), data collection (M.F., B.C., J.H., T.R., F.K., A.W., G.O., R.O., A.P.), performed sc-RNA sequencing and analyzed data (S.Sa., G.S., G.U., A.Mai., M.T., S.So., M.K.), performed flow cytometry analysis and analyzed data (S.E., M.F., T.H., S.So.), performed Luminex and analyzed data (M.F., T.H.), pathology and microscopic analyses (A.Mar., P.O., B.Z.), visualization (S.Sa., M.F., G.U., A.Mar., S.D., D.Ö., Z.T., J.T.), wrote original draft (T.H., S.Sa., M.F.), review & final editing of the article (all). O.R., P.A., W.D., and S.S. contributed equally as senior authors.

## Competing interests

The authors declare no competing interests.
