## [Peer Review File · Nature Communications]

Immune cell dynamics deconvoluted by single-cell RNA sequencing in normothermic machine perfusion of the liverREVIEWER COMMENTS

Reviewer #1 (Remarks to the Author):

The manuscript by Hautz et al reports on the very detailed analysis on the composition and the dynamics of leukocytes in the liver during NMP. The authors have invested enormous energy into characterizing single immune cell populations from biopsy material using single cell RNAseq and immunohistochemistry. They further characterized passenger leukocyte populations and investigated the concentrations of soluble mediators released from immune cells. This analysis is performed in a time lapse fashion nicely illustrating the dynamics of the changes in leukocyte populations in the liver and their activation over time. The amount of data presented is enormous and can certainly serve as a very valuable resource for future studies focussing on NMP and its effect on liver transplants.

While the manuscript provides ample amounts of data and provides data on changes in Kupffer cells and neutrophils, several issues need attention.

- the authors refer to their comprehensive data analysis as resource for further investigation. Other data on single cell RNAseq analysis of immune cells from liver tissue have been published in the past (Zhao et al. 2020; Aizarani et al. Nature 2019, among others). To fully understand the data presented here, it would be necessary to compare them to the already published data and to obtain an understanding whether there are differences between the different immune cell populations at the start of the NMP.

- since Nat Communications aims at a broad readership it is necessary to explain the technical setting of NMP in more detail and not simply refer other publications. This is necessary for the reader to understand the importance of studying immune cells from biopsies and from eluate, and to understand whether immune cell populations repeatedly circulate through the organ during NMP.

- despite the enormous wealth of information from scRNAseq, the authors mainly refer to lists of genes that they investigated. Their finding of the dynamics of the changes in neutrophil and Kupffer cell numbers during NMP as well as the increase in IL-6 protein expression leads to the conclusion that there is ongoing inflammation through these two cell populations. Thus, the authors do not fully deploy the enormous information they have gathered to characterize the state of immune cell activation. The manuscript would greatly profit from further analysis of the scRNAseq data beyond UMAP analysis, for instance using GSEA or transcription factor activity analysis in particular immune cell populations. Furtheron, since the authors infer to crosstalk among immune cells in the liver, it would be very instructive to use bioinformatic tools such as CellPhoneDB to identify such potential cross-talk. At present, the bulk of the data is presented without in-depth analysis following the authors hypothesis that inflammation is locally happening in the liver

- along this line, it would be good if the authors clarified what they mean by liver-resident neutrophils. Moreover, liver macrophages are a very diverse cell population and the term "Kupffer cell" used by the authors seems to be outdated given the in-depth information they have accumulated. A more thorough analysis of the different monocyte/macrophage populations at the beginning and during NMP would greatly profit the manuscript.

- The authors mention that their results pave the way to better understand liver graft integrity. This is a very promising idea, since inflammatory mediators may evoke different forms of liver damage depending on critical parameters. Data from liver sinusoidal endothelial cells and hepatocytes could be employed to gain insight into how NMP affected them. This could also provide a very good link to changes in the immune cell populations and how they may cause damage to the liver

Reviewer #2 (Remarks to the Author):

This is an ambitious study which I recognise involved acquisition and analysis of large amounts of data

on large numbers of samples. The authors should be congratulated on using a range of sophisticated technologies very competently. The study provides proof of concept that such approaches could yield useful information about the functional integrity and inflammatory status of quite good quality livers since they were all accepted with intention to transplant. Very few comprehensive NMP studies using these approaches exist in the literature but many perfusion research groups are working along similar lines.

The strength of the manuscript resides in the demonstration of the proof of concept and feasibility that the technologies used can be applied to studying some of the aspects liver responses during the highly successful clinical application of NMP in liver transplantation.

My concerns and questions are as follows :-

1. The sample size and the outcomes in terms of donor livers transplanted and complications were very similar to the VITTAL clinical transplant study published in Nature Comms by Mergental and colleagues 2020 11,1,2939. This study I could not see cited which was quite surprising given its high profile. Minor point
2. Whilst the authors discussed beneficial (immunosuppressive/anti inflammatory, reparative) effects of NMP on the liver and different isolated cell phenotypes compared with potential detrimental pro inflammatory effects, it does little to define anything at a mechanistic level. As a consequence I see this as a manuscript containing very interesting but very preliminary data which diminishes its impact considerably.
3. as a consequence of 2 it is not possible to use this data to inform on potential therapeutic intervention strategies with any certainty whether they would be beneficial or not.
4. It is logical to assume that depletion of any tissue /liver resident inflammatory cells due to flushout during NMP would help to ameliorate tissue inflammation. However activated phagocytic cells are also likely to be required for removal of dead tissue and cells as part of the repair process and to maintain a sufficiently functional innate immune response to protect against infectious insults etc.
5. linked to 4 - I find it difficult to understand the relevance of the functional state of the perfusate inflammatory cells unless they somehow contribute to (exogenous) soluble mediators of inflammation to which the perfused liver could be exposed.
6. What is the fate of the ICs in the perfusate? do they remain in the perfusate throughout the perfusion or could they transmigrate back to the tissues. Ingress and egress of inflammatory cells form part of normal as well as pathological immune and inflammatory responses.
7. Selecting 4/30 perfusions to undertake the scRNAseq analysis seems woefully unrepresentative. Why was this number so small and it would also appear that they all came from the transplantable group. This needs much more granularity and justification in my opinion.

Reviewer #3 (Remarks to the Author):

This work investigates changes in cell composition and phenotype in human livers during normothermic machine perfusion. The knowledge derived from such a study should help with further optimization of this preservation technology and presumably contribute to improving patient outcomes. For this, the authors rely primarily on single cell RNAseq technology. This and the downstream investigations conducted to validate initial findings are significant strengths. Yet this manuscript would likely benefit from clearer positioning and improved flow/message.

Of course, ultimately it should be the author's prerogative to decide how results should be presented, also the points below are, for the most part, merely suggestions:

– Positioning: The authors tend at times to put forward technological achievements (“providing a single-cell atlas of the human liver that maps the dynamics...” over the scientific ones (advancing knowledge: what did we learn?). Presenting original research, including novel biological knowledge and insights should in my opinion take precedent. Else the authors take the risk of reducing the novelty of the work to merely having used a technology (that is now widely available) in a new biological/clinical context. There is undoubtedly value in that but in the present case, given the rather extensive validation and downstream investigative work this could squarely be presented as a research paper. And arguably, the work may also come short as a reference dataset/atlas, given the relatively modest number of organs profiled, which might not permit to capture the full breadth of inter-individual variation (and indeed, the discussion could include points about potential limitations of the study).

– Background/introduction: in my opinion is too cursory. Readers who are not familiar with this field will probably not be able to fully grasp the importance of the work. For instance, when and why NMP has been introduced and in what sense has it been a game changer? What do we already know about immunopathophysiology in this setting? What else are we trying to learn, etc...? These points tend to come through in the discussion but should really be made in the introduction.

– Results: the flow of the manuscript would benefit from a few sentences at the beginning and end of each paragraph that would help tie everything together. Just to summarize the main points and provide a rationale for what comes next.

– What did we learn? Given the amount of data generated it is easy to fill in a manuscript with description of the analyses and results! This is a lot of information but how much of this confirms earlier findings and how much of it is novel? What did we actually learn from these data? It is not altogether clear from reading the manuscript, possibly again in part because the use of this technology in this context is considered to be novel enough to justify publication. This is maybe best reflected in the conclusion, which points to "providing a comprehensive transcriptomic landscape of the immune cell repertoire of human livers at single cell resolution" as the main achievement (although the novelty here is that it was obtained in the context of NMP). How this information has contributed to advancement of scientific knowledge should be made more clear (especially since other observations mentioned in the conclusion could have seemingly made just as well by comprehensive flow cytometry)/

– And to this last point, regarding novelty of the biological findings, citation and discussion of some relevant published work is missing:

o <https://pubmed.ncbi.nlm.nih.gov/30561835/> Hepatology. 2019 Aug;70(2):682-695. doi:

10.1002/hep.30475. Epub 2019 Mar 13. Jassem et al. "...compared 12 NMP with 27 CS-preserved livers by performing gene microarray, immunoprofiling of hepatic lymphocytes, and immunochemistry staining of liver tissues for assessing necrosis, platelet deposition, and neutrophil infiltration, and the status of steatosis after NMP or CS prereperfusion and postreperfusion"

o <https://pubmed.ncbi.nlm.nih.gov/34730028/>. Am J Physiol Gastrointest Liver Physiol. 2022 Jan 1;322(1):G21-G33. Ohman et al. applied transcriptomic profiling and protein analysis to evaluate temporal changes in gene expression during NMP between functional and nonfunctional livers. They describe a robust activation within hours of innate immunity followed by activation of repair and homeostasis mechanisms.

Although earlier points might be mere recommendations, this one is of concern and should be addressed in a revised version of the manuscript and letter to the editor.

– On another, note: data would customarily already be deposited in GEO at the submission stage. The dataset can be kept private until the paper is accepted but reviewer are usually provided with a token to be able to access the record and verify that all relevant information has been provided.

– Figures and analyses are of good quality overall, but statistics are not always provided on some of the plots (e.g. violin or line plots). Some of the text on Figures 6, 7 and 8 is barely legible.

Point-by-point response

Resubmission: Hautz T et al: “*Immune cell dynamics deconvoluted by single-cell RNA sequencing in normothermic machine perfusion of the liver*”

Reviewer #1 (Remarks to the Author):

The manuscript by Hautz et al reports on the very detailed analysis on the composition and the dynamics of leukocytes in the liver during NMP. The authors have invested enormous energy into characterizing single immune cell populations from biopsy material using single cell RNAseq and immunohistochemistry. They further characterized passenger leukocyte populations and investigated the concentrations of soluble mediators released from immune cells. This analysis is performed in a time lapse fashion nicely illustrating the dynamics of the changes in leukocyte populations in the liver and their activation over time. The amount of data presented is enormous and can certainly serve as a very valuable resource for future studies focussing on NMP and its effect on liver transplants.

Response: We thank the reviewer for the positive feedback on our manuscript recognizing the resource value of the generated datasets.

While the manuscript provides ample amounts of data and provides data on changes in Kupffer cells and neutrophils, several issues need attention.

- the authors refer to their comprehensive data analysis as resource for further investigation. Other data on single cell RNAseq analysis of immune cells from liver tissue have been published in the past (Zhao et al. 2020; Aizarani et al. Nature 2019, among others). To fully understand the data presented here, it would be necessary to compare them to the already published data and to obtain an understanding whether there are differences between the different immune cell populations at the start of the NMP.

Response: Thank you for pointing out this important aspect. We fully agree that our results should be discussed in the context of already available scRNASeq datasets of the human liver. Most importantly, we identified neutrophils as the most prevalent immune cell population. To best of our knowledge, the neutrophil lineage is entirely dismissed in hitherto published scRNASeq datasets of human livers. As discussed in the manuscript this discrepancy is most likely attributed to methodological pitfalls rather than to a biological phenomenon. Because neutrophils are a remarkably short-lived (circulatory half-life of 7-10 hours in humans 8), extremely fragile cell type that is particularly sensitive to tissue dissociation, sorting, or freezing, a quick and yet gentle workflow from tissue dissociation to cell lysis is essential to preserve these cells. Hence, the detection of neutrophils by scRNASeq is hampered from the outset in studies using frozen tissue or in studies where certain cell types were enriched by FACS (e.g. Aizarani et al. 2019 ¹; Zheng et al. 2017 ²; Zhang et al. 2019 ³; Tamburini et al. 2019 ⁴). Moreover, neutrophils express an exceptionally low amount of mRNA molecules (see also Fig 2H), which impedes their recovery in scRNASeq data. We could recently demonstrate that neutrophils cannot be appropriately detected especially in datasets generated with the droplet-based 10x Chromium platform and only to a very limited extent when applying other platforms ⁵. Of note, many of hitherto published liver scRNASeq studies were conducted using the 10x platform (e.g. MacParland et al. 2018 ⁶, Zhao et al. 2020 ⁷, Ramachandran et al. 2019 ⁸). We applied the microwell-based BD Rhapsody scRNASeq platform which allows to capture

relatively high numbers of mRNA molecules per cell and thus enhances the recovery of low-mRNA-content cells according to benchmarking of high-throughput scRNASeq platforms⁵. Recently, our group utilized this platform in lung cancer tissues to characterize the neutrophil lineage and to describe neutrophil subclusters based on identified marker genes⁵. Hence, using freshly isolated biopsies to perform scRNASeq of cells with low-mRNA content using the BD Rhapsody platform allowed us for the first time to describe neutrophils to such an extent in the human liver.

A paragraph pointing out the specificity of the sequencing platform used in this study to recover neutrophils by scRNASeq has been provided in the discussion section of our originally submitted manuscript. This paragraph has now been extended and supplemented with data from other studies performing scRNASeq analysis (of immune cells) from liver tissue as suggested by the reviewer. For details, please see page 17 of the discussion section.

- since *Nat Communications* aims at a broad readership it is necessary to explain the technical setting of NMP in more detail and not simply refer other publications. This is necessary for the reader to understand the importance of studying immune cells from biopsies and from eluate, and to understand whether immune cells repeatedly circulate through the organ during NMP.

Response: Thank you for your valuable comment. Details on the technical settings of liver NMP have been added to the introduction section (pages 4-5) of the revised manuscript and a language appropriate for the broad *Nat Communications* readership has now been used. We hope that this helps to better indicate, why studying immune cell populations and mobilization during *ex vivo* organ perfusion is of importance. The relevant literature has now been cited.

- despite the enormous wealth of information from scRNAseq, the authors mainly refer to lists of genes that they investigated. Their finding of the dynamics of the changes in neutrophile and Kupffer cell numbers during NMP as well as the increase in IL-6 protein expression leads to the conclusion that there is ongoing inflammation through these two cell populations. Thus, the authors do not fully deploy the enormous information they have gathered to characterize the state of immune cell activation. The manuscript would greatly profit from further analysis of the scRNAseq data beyond UMAP analysis, for instance using GSEA or transcription factor activity analysis in particular immune cell populations. Furtheron, since the authors infer to crosstalk among immune cells in the liver, it would be very instructive to use bioinformatic tools such as db cellphone to identify such potential cross-talk. At present, the bulk of the data is presented without in-depth analysis following the authors hypothesis that inflammation is locally happening in the liver

Response: We fully agree with the reviewer. To better characterize the state of immune cell activation and to depict cellular crosstalk between hepatic immune and parenchymal cells during liver NMP, adequate bioinformatic tools were now applied to analyze the scRNASeq data set. Specifically, GSEA, transcription factor activity analysis as well as cell2cell communication analysis were applied as suggested. The focus of the analysis was put on the dynamics during NMP [timepoint T0 (pre NMP) vs. T1 (end of NMP)]. The new findings are summarized in the results, main figures 4-6 and suppl. file of our revised manuscript. We hope that the more comprehensive and advanced bioinformatic work-up provides a more detailed and meaningful picture of how the immune cell dynamics of human livers are affected during *ex vivo* NMP. Reflections on the novel findings have been added to the discussion section (pages 17-20) of the revised manuscript.

- along this line, it would be good if the authors clarified what they mean by liver-resident neutrophils. Moreover, liver macrophages are a very diverse cell population and the term "Kupffer cell" used by the authors seems to be outdated given the in-depth information they have accumulated. A more thorough analysis of the different monocyte/macrophage populations at the beginning and during NMP would greatly profit the manuscript.

Response: Thank you for your valuable comment. Donor livers were thoroughly flushed with perfusion solution prior to biopsy sampling and connection to the perfusion device. Moreover, packed red cell bags which are used as perfusion solution are leukocyte-depleted (30 Gy irradiation). According to local standards, one packed unit of leukocyte-depleted red cell bag contains a maximum of 1×10^6 leukocytes. In the light of these conditions, we would consider "liver-resident" immune cells as inclusive of all types of immune cells in the donor liver as well as the perfusate (as per flow cytometry analysis). To avoid confusions, however, "liver-resident" has been deleted throughout the manuscript for the description of immune cell types such as neutrophils.

We agree with the reviewer that liver macrophages are an important major and diverse immune cell population in the liver, consisting of a variety of different subtypes beyond Kupffer cells. Hence, we have annotated an immune cell lineage as "monocytes/macrophages" according to the reviewer's suggestion and now also performed subcluster analysis to characterize distinctive subpopulation according to specific marker genes (e.g. from MacParland S et al., Nature communications 2018 ⁶). For details, please see the corresponding results referring to the monocytes/macrophages lineage (pages 11-12).

Moreover, subpopulations of the neutrophils as the largest intrahepatic immune cell population were annotated accordingly. Remarkably, the N2 cluster – which was found enriched during NMP – closely resembled an aged/chronically activated/exhausted phenotype recently identified also in human lung cancer tissues ⁵. These novel findings have now been integrated and discussed in the revised manuscript version.

- The authors mention that their results pave the way to better understand liver graft integrity. This is a very promising idea, since inflammatory mediators may evoke different forms of liver damage depending on critical parameters. Data from liver sinusoidal endothelial cells and hepatocytes could be employed to gain insight into how NMP affected them. This could also provide a very good link to changes in the immune cell populations and how they may cause damage to the liver

Response: Thank you for pointing out this significant and important aspect. To address this question, we performed additional analysis and have added new findings to the manuscript. DEG analysis revealed, that NMP induces a pro-angiogenic switch in neutrophils indicated by VEGFA as ranking amongst the top up-regulated genes at the end of NMP (Fig. S6C). The added cell-2-cell communication analysis revealed markedly elevated VEGFA-KDR1/FLT1 signaling from neutrophils towards endothelial cells (see Fig. 6A). These new findings suggest, that endothelial cells are activated during NMP and that NMP may induce re-vascularization of damaged tissue.

Further to this, we found that one distinctive monocyte/macrophage subcluster (M3) expressed high levels of FN1 (fibronectin) known to contribute to extracellular matrix formation and wound repair. Cell2cell communication analysis in monocytes/macrophages revealed strongly elevated FN1 signaling towards its cognate receptor ITGB1 expressed on cholangiocytes.

Since the focus of this study was to evaluate the dynamic changes of liver-resident inflammatory cells during liver NMP, we performed scRNASeq of cells with low mRNA content employing

the BD Rhapsody™ scRNASeq platform. This approach, however, may not capture the entire extent of parenchymal cells/hepatocytes. Hepatocyte gene profiling needs to be interpreted in light of this methodological limitations. Future studies may help to further elucidate the impact of immune-cell mediated tissue damage or repair on hepatocytes during NMP.

Reviewer #2 (Remarks to the Author):

This is an ambitious study which I recognise involved acquisition and analysis of large amounts of data on large numbers of samples. The authors should be congratulated on using a range of sophisticated technologies very competently. The study provides proof of concept that such approaches could yield useful information about the functional integrity and inflammatory status of quite good quality livers since they were all accepted with intention to transplant. Very few comprehensive NMP studies using these approaches exist in the literature but many perfusion research groups are working along similar lines. The strength of the manuscript resides in the demonstration of the proof of concept and feasibility that the technologies used can be applied to studying some of the aspects liver responses during the highly successful clinical application of NMP in liver transplantation.

Response: We highly appreciate the encouraging comment of the reviewer.

My concerns and questions are as follows :-

1. The sample size and the outcomes in terms of donor livers transplanted and complications were very similar to the VITTAL clinical transplant study published in Nature Comms by Mergental and colleagues 2020 11,1,2939. This study I could not see cited which was quite surprising given its high profile. Minor point

Response: Thank you for this comment. The study is now cited according to the reviewer suggestion.

2. Whilst the authors discussed beneficial (immunosuppressive/anti-inflammatory, reparative) effects of NMP on the liver and different isolated cell phenotypes compared with potential detrimental pro-inflammatory effects, it does little to define anything at a mechanistic level. As a consequence I see this as a manuscript containing very interesting but very preliminary data which diminishes its impact considerably.

Response: We agree that subsequent studies should address functional and mechanistic aspects of immunomodulation and regeneration in human livers during NMP. For this first comprehensive assessment of the cellular gene expression profiles, the trafficking behavior of immune cells and their phenotypes were investigated to provide a valuable resource also for other researchers in the field. For the revision of this manuscript, advanced bioinformatic tools were introduced in order to better address cell-to-cell communication aspects. While this does not fully substitute for mechanistic studies, a more precise picture of the immune status of a human donor liver during NMP was achieved. We are confident that the data obtained and the assessment performed in this study will set the stage for many functional and mechanistic investigations, as it sets a valuable groundwork for any future interventional studies. We hope that the reviewer agrees that our efforts markedly improved the impact of our work and that the data is of value for the liver immunology/liver transplantation community.

3. As a consequence of 2 it is not possible to use this data to inform on potential therapeutic intervention strategies with any certainty whether they would be beneficial or not.

Response: We agree with this statement. Based on our findings we can speculate, that blocking CXCR1/2 during neutrophil activation or specific leukocyte elimination strategies may represent potential ideas for intervention with the aim to avoid excessive immune activation during liver NMP. Additional bioinformatic analyses of our scRNASeq data (transcription factor activity and cell crosstalk during liver NMP) fuel the hypothesis, that blocking the CXCR1/2-IL-8 axis may be beneficial in liver NMP. However, any such claims remain purely speculative at this point and all such hypotheses require validation in future pre-clinical and clinical interventional studies.

4. It is logical to assume that depletion of any tissue /liver resident inflammatory cells due to flushout during NMP would help to ameliorate tissue inflammation. However, activated phagocytic cells are also likely to be required for removal of dead tissue and cells as part of the repair process and to maintain a sufficiently functional innate immune response to protect against infectious insults etc.

Response: We fully agree with the consideration. Subsets of neutrophils and macrophages are critically important for healing, regeneration and immune modulation. Unspecific filtration and immune cell depletion/wash out during liver NMP may thus induce undesired and potentially negative effects. A respective statement has been added in the discussion. In order to further deepen this aspect, additional analyses of neutrophil and monocyte/macrophage subsets with regenerative/tolerogenic capacity have now been performed. An upregulation of these cell populations was found towards the end of NMP.

5. linked to 4 - I find it difficult to understand the relevance of the functional state of the perfusate inflammatory cells unless they somehow contribute to (exogenous) soluble mediators of inflammation to which the perfused liver could be exposed.

Response: We agree with the reviewer that the biologic significance of circulating perfusate immune cells remains to be further elucidated. This limitation was stated in our revision (page 19). In this study we quantified immune cell release and dynamics in the perfusate over a course of 24 h of human liver NMP. We feel that the comprehensive analysis of the cellular composition, cell trafficking and the dynamics of the respective cell phenotype and the cytokine profile of the perfusate amalgamate to a meaningful picture of the inflammatory processes during NMP. Employing advance bioinformatic techniques, we further aimed to obtain a precise picture between the crosstalk of tissue and perfusate immune cells during NMP. This may now serve as the foundation for interventional studies aiming at clarification of the individual components and mechanisms of inflammation in this environment.

6. What is the fate of the ICs in the perfusate? do they remain in the perfusate throughout the perfusion or could they transmigrate back to the tissues. Ingress and egress of inflammatory cells form part of normal as well as pathological immune and inflammatory responses.

Response: Thank you for pointing this out. We have addressed this aspect in the discussion.

7. Selecting 4/30 perfusions to undertake the scRNAseq analysis seems woefully unrepresentative. Why was this number so small and it would also appear that they all came

from the transplantable group. This needs much more granularity and justification in my opinion.

Response: We have now significantly extended the NMP liver atlas. To increase granularity, we performed scRNASeq as well as FCM analysis of 4 additional livers. Collectively, the study now contains scRNASeq data from 22 biopsies taken from eight human livers. The sample size increased from 56,560 cells to 118,448 cells and the number of neutrophils increased from 18,379 to 57,564. For details, please see the updated results and revised Figures 2-6 as well as the respective supplementary file. The four additional livers added to the study were selected based on the donor criteria fitting the extended criteria profile for liver grafts. Since these organs were inferior in quality compared to the first set of four livers, the assessment was critically enhanced by adding the aspect of standard versus extended criteria organs. From the additional livers included in the study, two were transplanted while two were discarded after NMP. This is consistent with their quality profile and the fact that such marginal livers show a higher discard rate. (see also VITTAL trial, Mergental et al, Nat Commun 2020)⁹. The neutrophil and monocytes/macrophage cytokine expression profiles assessed on a transcriptomic level was compared between transplanted and discarded livers. Data on protein levels of cytokines in the perfusate of transplanted vs discarded livers is provided in Figure 8 and the Supplementary file.

Reviewer #3 (Remarks to the Author):

This work investigates changes in cell composition and phenotype in human livers during normothermic machine perfusion. The knowledge derived from such a study should help with further optimization of this preservation technology and presumably contribute to improving patient outcomes. For this, the authors rely primarily on single cell RNAseq technology. This and the downstream investigations conducted to validate initial findings are significant strengths. Yet this manuscript would likely benefit from clearer positioning and improved flow/message.

Response: We highly appreciate the positive feedback on our manuscript and thank the reviewer for his/her valuable and very constructive criticism.

Of course, ultimately it should be the author's prerogative to decide how results should be presented, also the points below are, for the most part, merely suggestions:

Ø Positioning: The authors tend at times to put forward technological achievements (“providing a single-cell atlas of the human liver that maps the dynamics...” over the scientific ones (advancing knowledge: what did we learn?). Presenting original research, including novel biological knowledge and insights should in my opinion take precedent. Else the authors take the risk of reducing the novelty of the work to merely having used a technology (that is now widely available) in a new biological/clinical context. There is undoubtedly value in that but in the present case, given the rather extensive validation and downstream investigative work this could squarely be presented as a research paper. And arguably, the work may also come short as a reference dataset/atlas, given the relatively modest number of organs profiled, which might not permit to capture the full breadth of inter-individual variation (and indeed, the discussion could include points about potential limitations of the study).

Response: Thank you for pointing this out. We have revised our manuscript in order to put a stronger focus on our findings rather than the methods applied. In that spirit, we have further enhanced the robustness of the assessment by increasing the number of livers in the analysis, enhancing the bioinformatic substance and investigating the aspect of standard criteria vs extended criteria (transplanted vs. discarded) livers.

Ø Background/introduction: in my opinion is too cursory. Readers who are not familiar with this field will probably not be able to fully grasp the importance of the work. For instance, when and why NMP has been introduced and in what sense has it been a game changer? What do we already know about immunopathophysiology in this setting? What else are we trying to learn, etc...? These points tend to come through in the discussion but should really be made in the introduction.

Response: Thank you for pointing this out. The current state of knowledge on liver NMP has now been appropriately and well-understandably summarized in the introduction. We have also addressed the immune-pathophysiology in the NMP setting, however, the very reason for the trial was that the current understanding is rather rudimentary. In this spirit, we lead from this statement to the targets and potential learning from the trial. The respective literature has been cited. We hope that you find these changes satisfactory.

Ø Results: the flow of the manuscript would benefit from a few sentences at the beginning and end of each paragraph that would help tie everything together. Just to summarize the main points and provide a rationale for what comes next.

Response: Thank you, your suggestion has been implemented.

Ø What did we learn? Given the amount of data generated it is easy to fill in a manuscript with description of the analyses and results! This is a lot of information but how much of this confirms earlier findings and how much of it is novel? What did we actually learn from these data? It is not altogether clear from reading the manuscript, possibly again in part because the use of this technology in this context is considered to be novel enough to justify publication. This is maybe best reflected in the conclusion, which points to "providing a comprehensive transcriptomic landscape of the immune cell repertoire of human livers at single cell resolution" as the main achievement (although the novelty here is that it was obtained in the context of NMP). How this information has contributed to advancement of scientific knowledge should be made more clear (especially since other observations mentioned in the conclusion could have seemingly made just as well by comprehensive flowcytometry)/

Response: We are very grateful for your supportive and helpful comments on how to improve the message and writing of our manuscript. We have now tried to emphasize potential learnings from our study (beyond the simple fact that it is a most comprehensive gene atlas of the liver), which we find meaningful and important for the advancement of the field: (1) the content, the phenotype and the dynamics of the inflammatory cells in liver and NMP (2) the corresponding cell2cell communication patterns pointing out to potential mechanisms. Any further conclusion we would draw from the study, we fear, would be too speculative as pointed out by reviewer 2. We hope you find this approach and compromise satisfactory.

Ø And to this last point, regarding novelty of the biological findings, citation and discussion of some relevant published work is missing:

o <https://pubmed.ncbi.nlm.nih.gov/30561835/> Hepatology. 2019 Aug;70(2):682-695. doi: 10.1002/hep.30475. Epub 2019 Mar 13. Jassem et al. "...compared 12 NMP with 27 CS-preserved livers by performing gene microarray, immunoprofiling of hepatic lymphocytes, and immunochemistry staining of liver tissues for assessing necrosis, platelet deposition, and neutrophil infiltration, and the status of steatosis after NMP or CS prereperfusion and postreperfusion"

o <https://pubmed.ncbi.nlm.nih.gov/34730028/>. Am J Physiol Gastrointest Liver Physiol. 2022 Jan 1;322(1):G21-G33. Ohman et al. applied transcriptomic profiling and protein analysis to evaluate temporal changes in gene expression during NMP between functional and nonfunctional livers. They describe a robust activation within hours of innate immunity followed by activation of repair and homeostasis mechanisms.

Response: Thank you. Both studies are now cited accordingly.

Although earlier points might be mere recommendations, this one is of concern and should be addressed in a revised version of the manuscript and letter to the editor.

Ø On another, note: data would customarily already be deposited in GEO at the submission stage. The dataset can be kept private until the paper is accepted but reviewer are usually provided with a token to be able to access the record and verify that all relevant information has been provided.

Response: scRNASeq data have been deposited in GEO (GSE216584) as suggested and reviewers have access to the original data. Sequence data that support the findings of this study (all Figures) is available through the NCBI GEO accession GSE216584. The source code to reproduce the data analysis is available from <https://github.com/icbi-lab/nmp-liver>. Processed input data and containerized software dependencies required to execute the code are available from zenodo: <https://doi.org/10.5281/zenodo.7249006>.

Ø Figures and analyses are of good quality overall, but statistics are not always provided on some of the plots (e.g. violin or line plots). Some of the text on Figures 6, 7 and 8 is barely legible.

Response: The missing statistics has been provided and the text on former figures 6, 7 and 8 (revised version Figures 7 and 8 as figure 8 of the original submitted version has been shifted to the suppl. figures) has been increased in size.

References

- 1 Aizarani, N. *et al.* A human liver cell atlas reveals heterogeneity and epithelial progenitors. *Nature* **572**, 199-204 (2019). <https://doi.org:10.1038/s41586-019-1373-2>
- 2 Zheng, C. *et al.* Landscape of Infiltrating T Cells in Liver Cancer Revealed by Single-Cell Sequencing. *Cell* **169**, 1342-1356.e1316 (2017). <https://doi.org:10.1016/j.cell.2017.05.035>
- 3 Zhang, Q. *et al.* Landscape and Dynamics of Single Immune Cells in Hepatocellular Carcinoma. *Cell* **179**, 829-845.e820 (2019). <https://doi.org:10.1016/j.cell.2019.10.003>
- 4 Tamburini, B. A. J. *et al.* Chronic Liver Disease in Humans Causes Expansion and Differentiation of Liver Lymphatic Endothelial Cells. *Front Immunol* **10**, 1036 (2019). <https://doi.org:10.3389/fimmu.2019.01036>
- 5 Salcher, S. *et al.* High-resolution single-cell atlas reveals diversity and plasticity of tissue-resident neutrophils in non-small cell lung cancer. *bioRxiv*, 2022.2005.2009.491204 (2022). <https://doi.org:10.1101/2022.05.09.491204>
- 6 MacParland, S. A. *et al.* Single cell RNA sequencing of human liver reveals distinct intrahepatic macrophage populations. *Nat Commun* **9**, 4383 (2018). <https://doi.org:10.1038/s41467-018-06318-7>
- 7 Zhao, J. *et al.* Single-cell RNA sequencing reveals the heterogeneity of liver-resident immune cells in human. *Cell Discov* **6**, 22 (2020). <https://doi.org:10.1038/s41421-020-0157-z>
- 8 Ramachandran, P. *et al.* Resolving the fibrotic niche of human liver cirrhosis at single-cell level. *Nature* **575**, 512-518 (2019). <https://doi.org:10.1038/s41586-019-1631-3>
- 9 Mergental, H. *et al.* Transplantation of discarded livers following viability testing with normothermic machine perfusion. *Nat Commun* **11**, 2939 (2020). <https://doi.org:10.1038/s41467-020-16251-3>

REVIEWERS' COMMENTS

Reviewer #3 (Remarks to the Author):

All points have been satisfactorily addressed. There are no further critiques/comments from my end.

Point-by-point response

Resubmission: Hautz T et al: *“Immune cell dynamics deconvoluted by single-cell RNA sequencing in normothermic machine perfusion of the liver”*

We highly appreciate the positive feedback on our manuscript and thank the reviewer for his/her valuable and very constructive criticism.

Stefan Schneeberger and Dominik Wolf on behalf of all co-authors.